# ADASPLASH-2: Faster Differentiable Sparse Attention

**Nuno M. T. Gonçalves** [* 1 2 3]  **Hugo Pitorro** [* 1 2]  **Vlad Niculae** [4]  **Edoardo M. Ponti** [5]  **Lei Li** [3]
**André F. T. Martins** [1 2 6 7]  **Marcos V. Treviso** [1 2 7]

## Abstract

Sparse attention has been proposed as a way to alleviate the quadratic cost of transformers, a central bottleneck in long-context training. A promising line of work is $\alpha$-entmax attention, a differentiable sparse alternative to softmax that enables input-dependent sparsity yet has lagged behind softmax due to the computational overhead necessary to compute the normalizer $\tau$. In this paper, we introduce ADASPLASH-2, which addresses this limitation through a novel histogram-based initialization that reduces the number of iterations needed to compute $\tau$ to typically 1–2. The key idea is to compute a coarse histogram of attention scores on the fly and store it in on-chip SRAM, yielding a more accurate initialization that enables fast forward and backward computation. Combined with a sparsity-aware GPU implementation that skips zero blocks with low overhead, ADASPLASH-2 matches or improves per-step training time relative to FlashAttention-2 when block sparsity is moderate-to-high (e.g., >60%), which often occurs at long-context lengths. On downstream tasks, models trained with our efficient $\alpha$-entmax attention match softmax baselines at short-context lengths and achieve substantial gains in long-context settings.

## 1. Introduction

The self-attention mechanism in transformers constitutes a major computational bottleneck since computing and materializing the score matrix $\boldsymbol{S} = \boldsymbol{Q}\boldsymbol{K}^\top$ incurs quadratic time and memory complexity, limiting scalability to

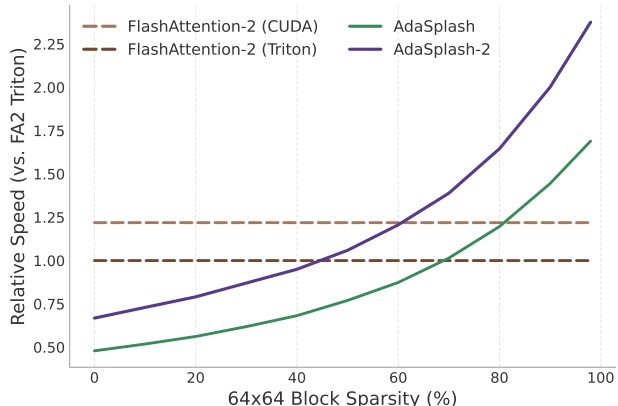

*Figure 1.* Runtime (forward + backward) as a function of input sparsity for causal attention. ADASPLASH-2, implemented in Triton, improves the sparsity-efficiency tradeoff, outperforming a highly-optimized CUDA version of FlashAttention-2 in moderate sparsity regimes and yielding larger gains at high block sparsity.

long sequences (Katharopoulos et al., 2020; Tay et al., 2022). FlashAttention (Dao et al., 2022; Dao, 2024; Shah et al., 2024) provides efficient GPU implementations that avoid this quadratic memory cost in the case of softmax attention. By making the computation IO-aware and by fusing operations over tiles that fit in on-chip fast memory, FlashAttention avoids storing intermediate scores and reduces memory complexity to linear in sequence length while achieving substantial speedups in training.

The key operation requiring efficient implementations is *normalization*. For a score vector $\boldsymbol{s}$, softmax can be written as

$$\mathrm{softmax}(\boldsymbol{s}) = \exp(\boldsymbol{s} - \tau\mathbf{1}), \quad \tau = \log\sum_j \exp(s_j), \quad (1)$$

where $\tau$ is an additive log-sum-exp normalizer. This form admits stable online accumulation and makes softmax particularly amenable to single-pass, tiled GPU implementations. FlashAttention and its successors (Dao, 2024; Shah et al., 2024) exploit precisely this structure.

However, softmax attention is inherently *dense* as every token receives nonzero probability mass. For long inputs, this density can be undesirable—attention mass spreads over many irrelevant tokens (Veličković et al., 2025), and token representations become less distinguishable (Barbero

*Equal contribution  [1]Instituto Superior Técnico, Universidade de Lisboa  [2]Instituto de Telecomunicações  [3]Carnegie Mellon University  [4]Language Technology Lab, University of Amsterdam  [5]University of Edinburgh  [6]TransPerfect  [7]ELLIS Unit Lisbon. Correspondence to: Nuno Gonçalves <nuno.m.goncalves@tecnico.ulisboa.pt>.

*Proceedings of the 43rd International Conference on Machine Learning*, Seoul, South Korea. PMLR 306, 2026. Copyright 2026 by the author(s).

et al., 2024)—motivating adaptive sparse alternatives. The *α-entmax transformation* (Peters et al., 2019) provides such an alternative: it is differentiable, yields exact zeros in an input-dependent way, and has been shown to improve long-context generalization relative to softmax (Vasylenko et al., 2025). Like softmax, $\alpha$-entmax also requires computing a normalizer similar to $\tau$ in (1) (to be described in §2.3); however, its evaluation is not additive and requires iterative root-finding methods. This has historically prevented $\alpha$-entmax attention from matching optimized softmax attention kernels in end-to-end training throughput.

While ADASPLASH (Gonçalves et al., 2025) has recently mitigated this problem by introducing a GPU-friendly implementation of $\alpha$-entmax attention, the provided algorithm still requires multiple expensive passes over the attention scores to compute $\tau$. This limits efficiency, particularly in the forward pass and in moderate-sparsity regimes where sparsity is present but not extreme.

In this work, we introduce ADASPLASH-2, a new and more efficient hardware-aware method for $\alpha$-entmax attention that reduces the computational cost of normalization. As we show in Figure 1, ADASPLASH-2 outperforms a highly-optimized CUDA implementation of FlashAttention-2 on moderate sparse regimes and can double the speedup at high sparsity cases. The key idea of our method is to construct a compact histogram of attention scores *on the fly* while streaming tiles through on-chip SRAM. This histogram yields a provable lower bound on the true normalizer $\tau$, providing a high-quality initialization that allows the subsequent root solver to converge to the *exact* solution in one (typically) or two iterations, without materializing dense intermediates. Our contributions include:[1]

- **Normalization via on-chip histogram.** We derive a provable lower and upper bound on the $\alpha$-entmax normalizer $\tau$ by computing a compact histogram of streamed scores entirely in on-chip SRAM, enabling fast $\alpha$-entmax evaluation without forming dense intermediates.

- **One-pass refinement with a safeguarded hybrid solver.** Initialized from the histogram estimate, a single additional pass typically suffices to recover the exact normalizer $\tau^\star$ using a safeguarded hybrid solver, yielding near-single-iteration convergence in practice for $\alpha \in \{1.5, 2.0\}$.

- **Efficient exploitation of dynamic sparsity on GPU.** We design an optimized attention kernel in Triton with fine-grained tiling and a lightweight bit-packed encoding of nonzero blocks, enabling input-dependent sparsity to be exploited with negligible overhead.

- **Strong empirical results on synthetic and language modeling benchmarks.** Our results show that

---

[1]Code: https://github.com/deep-spin/adasplash

ADASPLASH-2 is not only faster than FlashAttention-2 on moderate sparse regimes, but also matches or outperforms standard softmax attention on short- and long-context downstream tasks.

## 2. Background

### 2.1. Hardware Performance

Modern GPUs are built for efficient parallel execution over a hierarchical memory system. High-bandwidth memory (HBM) provides large capacity but higher access latency than the smaller, faster on-chip SRAM (Jia et al., 2018). High performance therefore depends on efficient use of SRAM to reduce bottlenecks from frequent HBM traffic. GPUs run computation as kernels launched over thousands of threads grouped into thread blocks; data are staged from HBM into SRAM for computation and then written back. Kernel fusion is a core optimization that merges multiple operations into a single kernel, avoiding intermediate HBM reads/writes by directly producing final outputs. While compilers such as `torch.compile` can automate fusion for relatively simple operator chains (Ansel et al., 2024), attention mechanisms typically require custom strategies to reorder operations and optimize memory usage effectively.

### 2.2. Standard Dot-Product Attention

Given a set of matrices $\boldsymbol{Q}, \boldsymbol{K}, \boldsymbol{V} \in \mathbb{R}^{n \times d}$ containing $d$-dimensional representations for $n$ queries, keys and values, the *dot-product self-attention* at a single head is computed in the following way (Vaswani et al., 2017):

$$\boldsymbol{S} = \frac{\boldsymbol{Q}\boldsymbol{K}^\top}{\sqrt{d}} \in \mathbb{R}^{n \times n}, \quad \boldsymbol{O} = \pi\left(\boldsymbol{S}\right)\boldsymbol{V} \in \mathbb{R}^{n \times d}. \quad (2)$$

The $\pi$ transformation maps rows to normalized probability vectors, with softmax (Equation 1) being the most common choice. Crucially, the normalization $\tau$ is a stable reduction along each row of $\boldsymbol{S}$ and can be computed online in a single pass (Milakov & Gimelshein, 2018).

**FlashAttention.** To address the costs of naive attention implementations, Dao et al. (2022) introduced FlashAttention, an algorithm that avoids the materialization of intermediate quadratic attention matrices via a GPU-aware implementation of online softmax (Milakov & Gimelshein, 2018), bringing the overall memory complexity to $\mathcal{O}(n)$. The key idea of FlashAttention is to split the inputs $\boldsymbol{Q}, \boldsymbol{K}, \boldsymbol{V}$ into blocks, load them from slow GPU high bandwidth memory (HBM) to the fast GPU on-chip SRAM, then compute the attention output regarding those blocks and, at the end, scale the output by the right normalization factor. Later, FlashAttention-2 (Dao, 2024) improved the original algorithm and effectively defined the algorithmic structure

adopted by subsequent variants, such as FlashAttention-3 (Shah et al., 2024), which introduced hardware-specific optimizations for NVIDIA Hopper GPUs (e.g., TMA, WG-MMA instructions, warp specialization) while preserving the same high-level algorithm.

### 2.3. $\alpha$-entmax Transformation

Softmax-based attention is inherently *dense*, as every key receives a strictly positive weight. A principled *differentiable* sparse alternative is the $\alpha$-entmax transformation (Peters et al., 2019). For $\alpha > 1$, $\alpha$-entmax can produce sparse probability vectors, and it interpolates between softmax ($\alpha \to 1$) and sparsemax ($\alpha = 2$, Martins & Astudillo 2016):

$$\alpha\text{-entmax}(\boldsymbol{s}) = [(\alpha - 1)\boldsymbol{s} - \tau\mathbf{1}]_+^{\frac{1}{\alpha-1}}, \qquad (3)$$

where $[\cdot]_+$ is the ReLU function, and $\tau \in \mathbb{R}$ is the (unique) normalizing constant which ensures the output is a valid probability distribution, $\sum_i \alpha\text{-entmax}(\boldsymbol{s})_i = 1$. This means that coordinates with $(\alpha - 1)s_i \le \tau$ become exactly zero. Importantly, $\alpha$-entmax yields dynamic sparsity, where the pattern of zeros depends on the input $\boldsymbol{s}$. Despite its flexibility, $\alpha$-entmax does not have a direct closed-form solution, so computing it requires more involved methods, which we discuss next.

### 2.4. $\alpha$-entmax Computation

Computing $\alpha$-entmax is equivalent to finding the normalizing constant $\tau^\star$ satisfying $f(\tau^\star) = 0$, where

$$f(\tau) := -1 + \sum_{j=1}^{n} [(\alpha - 1)s_j - \tau]_+^{\frac{1}{\alpha-1}}. \qquad (4)$$

The function $f$ is continuous and strictly decreasing in $\tau$, hence the root $\tau^\star$ is unique (Blondel et al., 2019).

**Sorting-based solvers.** For $\alpha = 1.5$ and $\alpha = 2$ (sparsemax), specialized solvers can be derived by sorting $\boldsymbol{s}$ and exploiting the fact that the support is given by the top-$k$ entries (Michelot, 1986; Duchi et al., 2008; Condat, 2016; Peters et al., 2019). These approaches are less attractive on GPU because sorting and support selection are expensive and hard to fuse.

**Bisection (bracketing).** For general $\alpha > 1$, $\tau^\star$ can be found by a bracketing method such as bisection (Blondel et al., 2019), which only needs an interval containing the root. Let $m = (\alpha - 1) \max(\boldsymbol{s})$. Following Peters et al. (2019), $\tau^\star$ satisfies

$$m - 1 \le \tau^\star \le m - n^{1-\alpha}, \qquad (5)$$

which provides a valid initial bracket. While bisection is robust, it converges linearly and requires repeatedly evaluating $f$, motivating faster refinement strategies on GPU.

**ADASPLASH.** Gonçalves et al. (2025) proposed a GPU-oriented implementation of $\alpha$-entmax attention that exploits dynamic sparsity and IO-awareness. Its main contribution is a hybrid *Halley-bisection* solver for the normalization threshold $\tau^\star$, combining fast local convergence with bracketing guarantees. Implemented as custom Triton (Tillet et al., 2019) kernels, ADASPLASH can skip zero blocks in forward and backward passes, improving performance at very high sparsity via a naive block masking strategy. However, refining $\tau^\star$ typically requires multiple iterations, each involving additional passes over the keys. In addition, its naive block masking strategy incurs extra overhead to determine which blocks can be skipped. Our method, described next, reduces the number of iterations and introduces a lightweight scheme to skip zero blocks.

## 3. Our Method

We present ADASPLASH-2, a hardware-aware sparse attention mechanism that achieves efficient $\tau$ computation through a histogram-based approximation built entirely in on-chip SRAM. Our approach improves runtime compared to ADASPLASH while maintaining the theoretical guarantees and sparsity benefits of $\alpha$-entmax.

### 3.1. Histogram Approximation

Given attention scores $\boldsymbol{s} \in \mathbb{R}^n$, our goal is to compute the threshold $\tau^\star$ such that Equation 4 satisfies $f(\tau^\star) = 0$. Since $\alpha$-entmax is invariant to adding a constant to all scores, we apply the change of variables

$$\boldsymbol{z} = (\alpha - 1)\boldsymbol{s} - (m - 1)\mathbf{1}, \qquad (6)$$

obtaining $\max(\boldsymbol{z}) = 1$. We henceforth work with centered scores $\boldsymbol{z}$, which restricts both the active entries and the threshold search to the unit interval (see Equation 5).

**Histogram construction.** With this change of variables, we discretize the interval $[0, 1]$ into $B$ equal-width bins with width $h = 1/B$. Formally, each score $z_j \in [0, 1]$ is mapped to a bin index as follows:

$$b(z_j) = \min(\lfloor Bz_j \rfloor, B - 1). \qquad (7)$$

At the same time we construct and maintain a histogram $\mathcal{H}$ in SRAM to store bin counts,

$$\mathcal{H}_k = \big|\{\, j : b(z_j) = k \,\}\big|, \qquad k = 0, \ldots, B - 1. \qquad (8)$$

Importantly, this representation only requires $\mathcal{O}(B)$ storage (independent of $n$), and entries with $z_j < 0$ are not included in the histogram since they cannot belong to the active set.

**Histogram objective.** Now in objective (4), we replace each score $z_j$ by the left endpoint of its bin. Concretely, if

$z_j$ is assigned to bin $k = b(z_j)$, we score it by $\tilde{z}_j := k/B$. This allows us to rewrite a discretized objective as a sum over bins:

$$
\begin{aligned}
f_h(\tau) &= -1 + \sum_{j=1}^{n} [\tilde{z}_j - \tau]_+^{\frac{1}{\alpha-1}} \\
&= -1 + \sum_{k=0}^{B-1} \sum_{j:b(z_j)=k} \left[\frac{k}{B} - \tau\right]_+^{\frac{1}{\alpha-1}} \\
&= -1 + \sum_{k=0}^{B-1} \mathcal{H}_k \cdot \left[\frac{k}{B} - \tau\right]_+^{\frac{1}{\alpha-1}}.
\end{aligned} \tag{9}
$$

Structurally, this construction replaces the original set of scores by $B$ distinct values with associated counts $H_k$, yielding a reduced problem of size $B \ll n$ with the same monotone structure as the exact normalization in Equation 4. In turn, this reduced problem can be solved using any algorithm developed for $\alpha$-entmax normalization. In particular, exact solvers apply for $\alpha = 1.5$ and $\alpha = 2$ (see §2.4). The key difference is that, in our setting, these solvers operate on a compact histogram representation that fits entirely in SRAM, minimizing the computation overhead compared to operating on the full set of $n$ scores, which need to be recomputed from HBM. For completeness, we provide the corresponding solver adaptations for the histogram objective in Appendix C.3. We now formalize the approximation quality of the histogram objective and establish guarantees on the accuracy of the resulting normalization.

## 3.2. Theoretical Properties and Guarantees

We first show that the histogram objective yields a conservative estimate of the true normalization threshold. We then characterize how the smoothness of $f$ and $\alpha$-entmax depends on $\alpha$, motivating when safeguarded higher-order updates (e.g., Halley (Scavo & Thoo, 1995)) are appropriate.

**Proposition 1** (Histogram Lower Bound). *Let $\alpha > 1$ and $h = 1/B$ be the bin width. Let $f$ and $f_h$ be defined as in Equations 4 and 9. If $\tau^\star$ denotes the unique root of $f(\tau) = 0$ and $\tau_h$ the unique root of $f_h(\tau) = 0$, then:*

$$
\tau^\star - h < \tau_h \leq \tau^\star. \tag{10}
$$

*Thus, the absolute error satisfies $0 \leq \tau^\star - \tau_h < h = \frac{1}{B}$.*

The proof is given in Appendix B.1. This result ensures that the histogram approximation never overestimates the true threshold ($\tau_h \leq \tau^\star$) and provides explicit control over the maximum approximation error through the bin width $h$. In particular, the error satisfies $0 \leq \tau^\star - \tau_h < h$, and the approximation converges to the true solution as $B \to \infty$.

Other binning approaches lead to different precision-sparsity tradeoffs. Using a *right-edge* strategy $b(z_j) = \lceil B \cdot z_j \rceil / B$

reverses the inequality to $\tau_h \geq \tau^\star$, yielding a sparser output at the cost of potentially discarding relevant scores, while a *centered* approach $b(z_j) = (\lfloor B \cdot z_j \rfloor + 0.5)/B$ may provide a more balanced approximation but does not induces any a one-sided bound guarantee. In ADASPLASH-2, we adopt the left-edge binning strategy to ensure that the active support induced by $\tau_h$ contains the true support.

**Smoothness and safe higher-order refinement.** Following our histogram-based threshold initialization, we typically apply a posterior refinement step that evaluates $f$ and its derivatives at candidate thresholds. Because each term $[z_j - \tau]_+^{1/\alpha-1}$ may be non-smooth at $\tau = z_j$, the existence and continuity of higher-order derivatives depend on $\alpha$. The next proposition makes this connection explicit, and we use it to justify when higher-order updates are well-behaved.[2]

**Proposition 2** (Continuity of $f$ and $\alpha$-entmax.). *Let $f(\tau)$ be defined as in Equation 4 for a given $\alpha$. If $1 < \alpha < \frac{t+1}{t}$, then $f \in \mathcal{C}^t$ and, consequentially, $\alpha$-entmax $\in \mathcal{C}^t$.*

The proof is given in Appendix B.2. Proposition 2 clarifies which higher-order updates are numerically safe. When $\alpha \in (1, 1.5]$, $f''$ is bounded, so Halley updates (which use $f''$) are numerically stable.[3] When $\alpha \in (1.5, 2]$, $f'$ remains well-behaved but $f''$ becomes unbounded, making Newton updates more robust than Halley in practice. When $\alpha > 2$, $f'$ also becomes unbounded, and thus gradient-based methods are less stable. Based on this, we propose a new safeguarded **hybrid solver**: we apply Halley steps for $\alpha \leq 1.5$, Newton steps for $1.5 < \alpha \leq 2$, and secant steps for $\alpha \geq 2$, always falling back to bisection whenever the proposed update falls outside the brackets. Combined with Proposition 1, this algorithm yields a refinement procedure that is both provably safe and fast in practice. We now outline the GPU-aware implementation of ADASPLASH-2.

## 3.3. GPU-Aware Implementation

Our implementation orchestrates multiple coordinated passes over the key matrix, and carefully exploits memory hierarchies for data movement between HBM and SRAM. Concretely, we perform a grid loop over $T_r = \lceil n/B_r \rceil$ query blocks, and for a specific $i^{\text{th}}$ query block $\boldsymbol{Q}_i \in \mathbb{R}^{B_r \times d}$ attending over key blocks $\{\boldsymbol{K}_j\}_{j=1}^{T_c}$ where $T_c = \lceil n/B_c \rceil$, our method proceeds as follows.

**Phase 1 (maximum).** Stream key blocks to compute the row-wise maximum $m_i$, which is scaled by $(\alpha - 1)$.

---

[2] We say $f \in \mathcal{C}^t$ if $f$ has continuous derivatives up to $t^{\text{th}}$ order.

[3] In fact from the monotonicity of $f''$ we have $|f''(\tau)| \leq n(2 - \alpha)/(\alpha - 1)^2$. This is an upper bound on the Lipschitz constant of $f'$, which by Kantorovich's theorem controls the interval of convergence for Newton's method (Kantorovich, 1949).

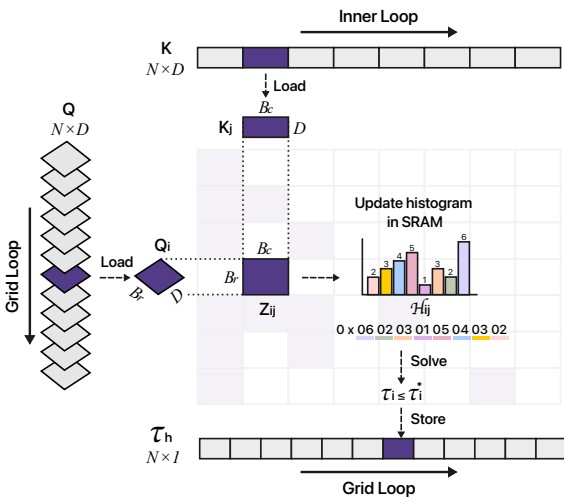

*Figure 2.* Diagram of the ADASPLASH-2 histogram kernel. For each query block $Q_i$ and key block $K_j$, we compute a score block $Z_{ij}$, update a per-row histogram $\mathcal{H}$ in SRAM, and use it to estimate an initial threshold $\tau_h$ for each query.

**Phase 2 (histogram).** Using $m_i$, we center and scale scores into $[0, 1]$ and build a $B$-bin per-row histogram entirely in SRAM while streaming over key blocks. To avoid atomics, we maintain local histogram accumulators for each position $(i, j)$ in a query-key block and reduce them across columns after all keys are processed. Each accumulator packs $B$ bins (representing bin counts) into a single $w$-bit integer ($b = w/B$ bits per bin). A score in bin $k$ updates its respective *local* accumulator via $\mathcal{H}^{\text{local}} \leftarrow \mathcal{H}^{\text{local}} + 2^{kb}$. Final bin counts are recovered using vectorized shift-and-mask operations with final reductions. Beyond fitting in SRAM, the histogram also provides an algorithmic speedup since binning implicitly orders scores by value. As a result, for $\alpha \in \{1.5, 2\}$, we can apply exact solvers directly to the bin counts and recover $\tau_h$ without an explicit sort step. The diagram in Figure 2 illustrates this process.

**Phase 3 (refinement).** Starting from $\tau_h$, we perform a fixed number of iterations to refine $\tau$ with a Hybrid solver that combines different root-finding methods (detailed in §3.2). In practice, we find that a single iteration is often sufficient for refinement due to the good accuracy of the histogram initialization. Simultaneously, we construct a binary block mask $\mathcal{M} \in \{0, 1\}^{T_r \times T_c}$ indicating which blocks contain non-zero attention weights. The mask is stored using bitpacking, with each group of 32 consecutive column blocks encoded in a single int32, requiring only $\mathcal{O}(T_r \times T_c)$ bits of memory. In contrast to ADASPLASH (Gonçalves et al., 2025), this design avoids auxiliary index buffers and incurs a negligible overhead, even for long contexts.

**Phase 4 (output).** Using mask $\mathcal{M}$, we traverse only nonzero blocks via GPU-native find-next-set (fns) instructions

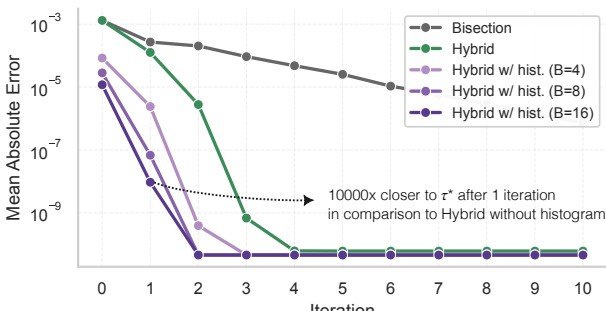

*Figure 3.* Comparison of mean absolute error of previous root-finding methods and our Hybrid approach with histogram initialization, measured against the exact solution for $\alpha = 1.5$.

to compute $O_i = \sum_{j:\mathcal{M}_{ij}=1} \alpha\text{-entmax}(Q_i K_j^\top, \tau) V_j$, enabling $\mathcal{O}(|\mathcal{M}|)$ traversal complexity rather than $\mathcal{O}(T_r \times T_c)$.

**Histogram capacity.** The bitpacking scheme naturally limits capacity since each bin can count up to $2^b - 1$ items before overflow. With $B_c$ parallel accumulators per query block, the maximum capacity is

$$C_{\max} = B_c \times (2^{w/B} - 1). \qquad (11)$$

For our default configuration ($w = 64$, $B = 8$, $B_c = 64$), this yields $C_{\max} = 16{,}320$ keys per query block, sufficient for sequences up to 16K tokens. When sequences exceed this capacity, we employ a periodic flushing of bin counts to a $B_c \times B$ accumulator sitting in SRAM to avoid overflow. We provide more information on this in Appendix C.5.

**Backward pass.** Crucially, gradients are nonzero only on the $\alpha$-entmax support (Peters et al., 2019), we therefore reuse the block mask obtained in the forward pass to efficiently iterate only over nonzero blocks when computing the gradients with respect to queries, keys, and values. For this, we make use of native GPU instructions like fns and popc (population count) for efficient traversal. Further implementation details are presented in §C and §F, including the complete forward and backward algorithms.

## 4. Experiments

Our goals are to (i) verify that our histogram-based initialization for $\tau$ reduces convergence steps; (ii) quantify the training-time speedups of ADASPLASH-2 relative to softmax-based baselines; and (iii) study model accuracy and long-context capabilities of LLMs trained with ADASPLASH-2.

### 4.1. Normalization Solver Analysis

Before end-to-end GPU benchmarks, we first isolate the normalization step and study how quickly different root-

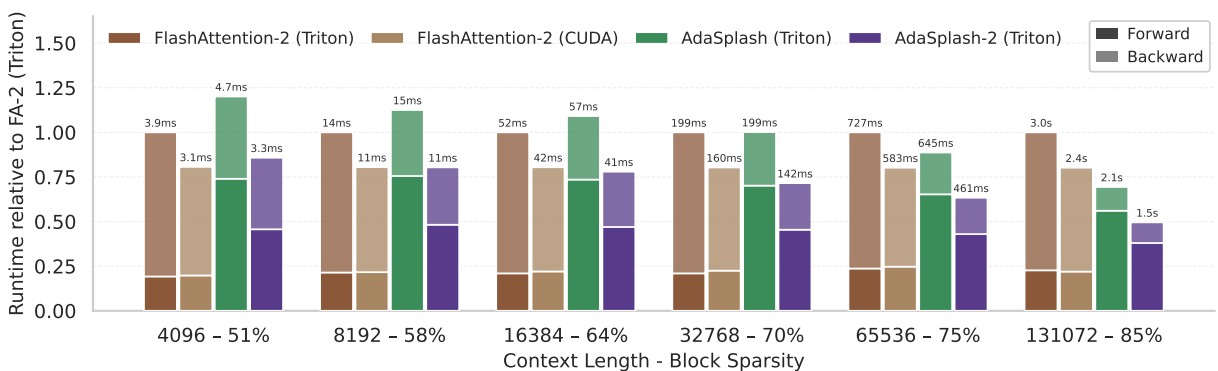

*Figure 4.* Runtime efficiency of causal self-attention implementations across context lengths of 4K-128K tokens with varying 64x64 block sparsity. Bar heights are normalized to FlashAttention-2 (Triton), with the opaque part denoting forward and the lighter part denoting backward. Numeric labels report the total forward + backward step time. Lower bars represent faster runtimes.

finding strategies recover the entmax threshold $\tau^\star$. Concretely, we follow Gonçalves et al. (2025) and sample a Gaussian score vector of length $n = 4096$ and compute a high-accuracy reference threshold $\tau^\star$ using an exact method (see §2.4). We then run (i) bisection; (ii) our safeguarded hybrid solver without histogram initialization;[4] and (iii) the same hybrid solver initialized with the histogram estimate $\tau_h$ using $B \in \{4, 8, 16\}$ bins. Figure 3 reports the mean absolute error averaged over 10 runs for $\alpha = 1.5$.

After the first iteration, the histogram initialization places the solver orders of magnitude closer to the true root, making subsequent refinement into a near one-pass correction. As expected, more bins lead to better initial estimates.

### 4.2. Efficiency Benchmark

We implement ADASPLASH-2, described in Section 3.1, in Triton. We compare it with FlashAttention-2 (Dao, 2024) implemented in CUDA and in Triton,[5] and ADAS-PLASH written in Triton (Gonçalves et al., 2025). A natural question is **why compare with FlashAttention-2 rather than FlashAttention-3**. Our efficiency experiments target NVIDIA Ampere GPUs, for which FlashAttention-2 is the reference implementation. FlashAttention-3 (Shah et al., 2024) introduces hardware-specific optimizations for NVIDIA Hopper GPUs (e.g., TMA, WGMMA instructions, warp specialization) while preserving the same high-level algorithm. This choice allows us to isolate algorithmic effects independent of such hardware tuning. We note that the low-level optimizations in FlashAttention-3 are largely orthogonal to ADASPLASH-2 and can be incorporated with additional engineering effort. All runtime benchmarks are performed in a single A6000 NVIDIA GPU.

---

[4]For $\alpha = 1.5$, our Hybrid solver empirically matches the Halley-Bisection solver of Gonçalves et al. (2025).

[5]https://github.com/triton-lang/triton/blob/main/python/tutorials/06-fused-attention.py

**Sparsity-speed tradeoff.** Figure 1 shows runtime as a function of block sparsity for random query, key and value tensors drawn from a zero-mean Gaussian. Following Gonçalves et al. (2025), we control the block sparsity of the attention probability matrix by changing the query variance, set head dimension to $d = 64$ and use bf16 precision. ADASPLASH-2 consistently outperforms ADAS-PLASH and, unlike FlashAttention, benefits from increased block sparsity, surpassing both Triton and CUDA implementations at moderate to high sparsity regimes. In the limit, ADASPLASH-2 achieves more than a $2\times$ speedup over both variants of FlashAttention-2. Under matched settings, ADASPLASH-2 is a strict improvement over AdaSplash-1, running faster at every sparsity level.

**Context scaling.** In this experiment, we do not hand-tune the input sparsity. Instead, we use the block-sparsity ratios from a fixed, highly-sparse layer of a 1B-parameter transformer language model at different sequence lengths (full results in the next subsection). Since ADASPLASH-2's runtime relative to FlashAttention-2 is driven mainly by block sparsity, a single representative layer suffices to isolate how the learned sparsity that emerges at long contexts translates into runtime; thus the results should be read as a learned-sparsity case study, not a whole-model throughput estimate. We describe how these ratios are extracted from our models and provide block sparsity heatmaps across sequence lengths in §E. Figure 4 reports the average per-step training runtime (forward + backward) as a function of context length, with bar heights normalized to FlashAttention-2 Triton (lower is better) and decomposed into forward and backward contributions.

As expected, ADASPLASH is slower than FlashAttention-2 in the forward pass due to the additional passes required to compute the normalization threshold $\tau$ (phases 1, 2, and 3 in §3.3). However, as context length increases, block sparsity naturally emerges, reducing the forward pass gap.

*Table 1.* RULER benchmark results for 1B parameter models trained up to 32K context-length. Best average results are in **bold**.

| Model | Len. | MK-1 | MK-2 | MK-3 | MQ | MV | NIAH-1 | NIAH-2 | NIAH-3 | CWE | FWE | VT | QA-H | QA-S | Avg. |
|---|---|---|---|---|---|---|---|---|---|---|---|---|---|---|---|
| Softmax (RoPE) | 4K | 80.0 | 62.2 | 38.2 | 47.6 | 43.4 | 100.0 | 100.0 | 98.8 | 5.1 | 24.9 | 8.0 | 24.8 | 35.0 | 51.4 |
| | 8K | 78.4 | 45.0 | 30.2 | 36.7 | 31.0 | 100.0 | 99.2 | 98.2 | 0.1 | 24.3 | 2.8 | 22.4 | 20.2 | 45.3 |
| | 16K | 67.6 | 10.0 | 14.8 | 22.1 | 22.7 | 100.0 | 99.2 | 99.2 | 0.0 | 19.0 | 0.2 | 17.0 | 18.7 | 37.7 |
| | 32K | 41.4 | 8.2 | 0.8 | 17.4 | 20.8 | 96.2 | 61.6 | 75.6 | 0.0 | 6.3 | 0.0 | 16.8 | 10.1 | 27.3 |
| Entmax (RoPE) | 4K | 75.4 | 26.0 | 23.6 | 29.9 | 52.3 | 100.0 | 100.0 | 98.8 | 36.3 | 32.3 | 3.9 | 25.8 | 36.7 | 49.3 |
| | 8K | 71.8 | 19.4 | 8.6 | 22.5 | 42.5 | 100.0 | 99.8 | 94.6 | 8.9 | 23.7 | 3.6 | 23.2 | 19.6 | 41.4 |
| | 16K | 53.2 | 8.6 | 1.0 | 12.9 | 22.0 | 100.0 | 81.4 | 79.4 | 0.1 | 18.6 | 1.2 | 18.2 | 20.8 | 32.1 |
| | 32K | 33.2 | 4.4 | 0.6 | 7.5 | 12.9 | 97.0 | 40.8 | 64.2 | 0.0 | 15.0 | 0.2 | 18.6 | 9.2 | 23.4 |
| Softmax (NAPE) | 4K | 88.2 | 29.8 | 13.0 | 28.9 | 27.9 | 100.0 | 100.0 | 100.0 | 35.3 | 27.4 | 14.8 | 21.8 | 37.9 | 48.1 |
| | 8K | 78.4 | 38.6 | 12.6 | 33.2 | 34.1 | 100.0 | 100.0 | 99.0 | 27.0 | 26.3 | 12.3 | 19.2 | 22.9 | 46.4 |
| | 16K | 85.6 | 19.6 | 7.4 | 26.8 | 30.6 | 100.0 | 100.0 | 98.8 | 18.0 | 25.6 | 9.8 | 20.0 | 21.3 | 43.3 |
| | 32K | 66.6 | 4.2 | 2.0 | 26.4 | 26.5 | 100.0 | 97.0 | 98.0 | 8.3 | 3.3 | 7.7 | 19.0 | 19.5 | 36.8 |
| Entmax (NAPE) | 4K | 81.2 | 12.2 | 27.2 | 51.9 | 55.6 | 100.0 | 100.0 | 79.8 | 48.9 | 58.1 | 31.4 | 30.2 | 34.5 | **54.7** |
| | 8K | 73.4 | 4.6 | 17.6 | 62.3 | 31.5 | 100.0 | 100.0 | 84.0 | 33.7 | 48.3 | 38.4 | 29.0 | 26.0 | **49.9** |
| | 16K | 79.2 | 1.4 | 9.8 | 38.7 | 24.0 | 100.0 | 100.0 | 74.2 | 22.9 | 51.1 | 34.5 | 27.2 | 27.5 | **45.4** |
| | 32K | 63.0 | 1.8 | 4.8 | 25.8 | 16.8 | 100.0 | 92.0 | 76.0 | 11.7 | 45.8 | 27.4 | 25.0 | 22.1 | **39.4** |

In the backward pass, even at shorter contexts, existing sparsity already yields faster runtimes for ADASPLASH-2, with the advantage increasing at longer contexts. Since the backward pass dominates training time, these gains are substantial. This suggests that as models scale to longer contexts, ADASPLASH-2 can translate naturally emerging sparsity into training efficiency gains, contrasting with the dense nature of softmax-based attention.

### 4.3. Language Modeling Benchmarks

To evaluate the effectiveness of $\alpha$-entmax attention in language modeling tasks, we train 350M and 1B parameters versions of the LLaMA-3 architecture (Grattafiori et al., 2024) using ADASPLASH-2 with $\alpha = 1.5$. Our experimental design focuses on general capabilities and long-context performance and, thus, has two distinct settings: (i) one version that does pretraining from scratch for 50B tokens from the DCLM-Edu dataset (Allal et al., 2025), and (ii) an alternative version with a context extension phase with the ProLong (Gao et al., 2025) methodology in the final 20% of tokens, endowing our models with 32K context length. Furthermore, following Vasylenko et al. (2025), who show that RoPE can be suboptimal for entmax attention, we also evaluate their proposed positional encoding scheme, NAPE: within each layer, half of the heads use no positional encoding (NoPE; Kazemnejad et al. 2023) and the other half use ALiBi (Press et al., 2022). We compare ADASPLASH-2 runs against softmax baselines with RoPE (Su et al., 2024) and with NAPE to ensure a fair comparison. Additional experimental details are provided in §D.

**Long-context results.** We start by investigating the performance of our models on the full RULER benchmark (Hsieh et al., 2024), evaluated up to 32K context length. The

*Table 2.* Results across context lengths for the In-Context Learning tasks in the HELMET benchmark. Best average result is in **bold**.

| Model | Len. | TREC-C | TREC-F | NLU | B77 | C150 | Avg. |
|---|---|---|---|---|---|---|---|
| Softmax (RoPE) | 8K | 62.4 | 33.6 | 12.6 | 6.0 | 13.6 | 25.6 |
| | 16K | 67.2 | 42.2 | 26.4 | 10.8 | 25.8 | 34.5 |
| | 32K | 69.2 | 47.8 | 25.8 | 12.8 | 35.6 | 38.2 |
| Entmax (RoPE) | 8K | 63.2 | 26.6 | 25.4 | 8.4 | 20.2 | 28.8 |
| | 16K | 66.4 | 35.8 | 32.8 | 13.8 | 27.4 | 35.2 |
| | 32K | 66.8 | 42.8 | 36.2 | 16.8 | 34.0 | 39.3 |
| Softmax (NAPE) | 8K | 58.8 | 30.8 | 27.2 | 13.4 | 32.6 | 32.6 |
| | 16K | 64.4 | 35.8 | 38.6 | 20.6 | 43.2 | 40.5 |
| | 32K | 71.2 | 45.6 | 41.4 | 21.2 | 51.0 | 46.1 |
| Entmax (NAPE) | 8K | 78.0 | 43.8 | 44.6 | 19.4 | 41.0 | **45.4** |
| | 16K | 83.4 | 56.4 | 58.0 | 25.0 | 57.0 | **56.0** |
| | 32K | 85.0 | 62.4 | 60.6 | 34.4 | 68.4 | **62.2** |

results are presented in Table 1. Overall, $\alpha$-entmax with NAPE achieves the strongest average performance across all sequence lengths, outperforming both softmax baselines. Among the different tasks, we note that the gains are particularly pronounced on Variable Tracking (*VT*)—a core subproblem underlying complex reasoning (Feng & Steinhardt, 2024; Dai et al., 2024)—as well as on common and frequent word extraction (*CWE* and *FWE*), which require precise aggregation of the relevant input spans.

Next, we evaluate the models' in-context learning (ICL) performance using the ICL subset from the HELMET benchmark (Yen et al., 2025), which measures few-shot performance across increasing context lengths. Table 2 shows that $\alpha$-entmax substantially improves ICL accuracy, especially when paired with NAPE, achieving the highest average score at every tested context length and outperforming softmax baselines. Notably, while all models improve as the context

*Table 3.* Short-context benchmark results for 350M and 1B models with 4K context length. Scores are computed with the OLMES framework using the *core_9mcqa* suite. Best perplexity and average results are in **bold**.

| Model | LMB (ppl) | LMB | ARC-E | ARC-C | CSQA | HS | OBQA | PIQA | SocialQA | WG | Avg. |
|---|---|---|---|---|---|---|---|---|---|---|---|
| *350M params.* | | | | | | | | | | | |
| Softmax with RoPE | 23.93 | 40.6 | 61.9 | 32.1 | 50.5 | 41.6 | 38.4 | 66.6 | 45.1 | 48.9 | 47.3 |
| Entmax with RoPE | 22.36 | 39.5 | 63.0 | 34.8 | 50.5 | 40.2 | 39.0 | 64.8 | 44.1 | 53.4 | 47.7 |
| Softmax with NAPE | 19.23 | 41.2 | 61.8 | 33.4 | 47.6 | 40.5 | 38.0 | 64.8 | 43.5 | 52.7 | 47.1 |
| Entmax with NAPE | **18.62** | 42.4 | 61.9 | 33.0 | 51.2 | 41.0 | 39.2 | 66.0 | 44.4 | 53.7 | **48.1** |
| *1B params.* | | | | | | | | | | | |
| Softmax with RoPE | 15.01 | 44.7 | 69.0 | 36.0 | 58.7 | 49.6 | 42.8 | 70.0 | 46.4 | 55.6 | 52.5 |
| Entmax with RoPE | 15.76 | 43.9 | 65.4 | 36.6 | 56.2 | 47.4 | 41.6 | 69.8 | 45.7 | 54.2 | 51.2 |
| Softmax with NAPE | 11.97 | 48.0 | 69.3 | 37.4 | 56.7 | 48.8 | 45.2 | 69.5 | 46.5 | 55.2 | 53.0 |
| Entmax with NAPE | **11.42** | 49.2 | 67.7 | 39.9 | 57.3 | 48.7 | 45.0 | 68.3 | 47.1 | 55.1 | **53.1** |

grows from 8K to 32K, we observe particularly large gains with $\alpha$-entmax + NAPE, suggesting that sparsity helps the model in leveraging the relevant in-context examples, reinforcing the finding that $\alpha$-entmax + NAPE constitutes a strong synergy for long-context modeling.

**Short-context results.** For evaluating short-context performance, we follow the few-shot prompting strategy under the OLMES evaluation standard (Gu et al., 2025). The results are shown in Table 3. We first observe a substantial perplexity gap between softmax and $\alpha$-entmax models on LAMBADA (*LMB*). Second, we note that $\alpha$-entmax models are on par or better across the board on accuracy-based downstream tasks at both model scales (350M and 1B params.). Taken together, these results highlight the strengths of sparse attention by not just outperforming dense models on long-context settings, but also on short-context tasks.

## 5. Related Works

$\alpha$**-entmax computation.** Exact computation exists for special cases such as $\alpha \in \{1.5, 2\}$ using sorting-based solvers (Duchi et al., 2008; Condat, 2016; Peters et al., 2019). For general $\alpha$, the normalization $\tau$ can be computed using root-finding procedures (Blondel et al., 2019). However, these approaches remain suboptimal due to slow convergence or reliance on complex data structures and sorting operations, which are difficult to optimize for hardware.

**Sparse attention mechanisms.** Sparse attention has been widely studied through *fixed* sparsity patterns (Zaheer et al., 2020; Beltagy et al., 2020). Recent methods aim for *data-dependent* sparsity, commonly via top-$k$ selection followed by softmax over the selected subset (Yuan et al., 2025; Liu et al., 2025; Nawrot et al., 2025). Top-$k$ is attractive for inference but can be unstable in training due to discontinuities. In contrast, $\alpha$-entmax induces exact, differentiable, input-adaptive sparse attention. ADASPLASH-2 focuses on

making $\alpha$-entmax attention efficient during training; while extending it to inference is non-trivial as the forward pass requires multiple key scans, and we leave a fully optimized inference kernel for future work.

**GPU-aware attention for training.** FlashAttention and its successors (Dao et al., 2022; Dao, 2024; Shah et al., 2024) provide highly optimized CUDA kernels for softmax attention via fused normalization and tiled accumulation that exploit GPU memory hierarchies. FlexAttention (Dong et al., 2024) offers a programmable interface for custom masks/score modifications while using similar fused, tiled execution. ADASPLASH (Gonçalves et al., 2025) offers a hardware-aware implementation of $\alpha$-entmax attention by solving $\tau$ with iterative refinement, but incurs considerable overhead due to repeated scans over keys and naive block-masking.

**Inference-time considerations.** ADASPLASH-2 targets training, where the forward-pass cost of computing $\tau$ is amortized by the sparsity-driven savings in the backward pass. Decoding is a distinct systems problem: it is memory-bound, and the relevant sparsity regime shifts from block-level to token-level—in our measurements, around 87/90/93/95% at 4/8/16/32K context for a 1B $\alpha$-entmax +NAPE model, well above the block sparsity exploited during training. A similar training/decoding kernel split exists even in the fully dense softmax case (FlashAttention vs. FlashDecoding). EntmaxKV (Duarte et al., 2026) is a direct complement at inference time, reusing the support induced by $\alpha$-entmax for support-aware decoding. We leave a dedicated decoding kernel for future work.

**Long-context modeling.** A growing body of work studies why transformer performance degrades as context length grows, attributing failures to the softmax function itself such as attention dispersion (Zhai et al., 2023; Veličković et al., 2025) and representational collapse (Barbero et al., 2024; Arroyo et al., 2025). Complementary lines improve

extrapolation through positional bias design (Press et al., 2022; Jelassi et al., 2024) and through length/entropy-aware scaling (Peng et al., 2024; Nakanishi, 2025; Zhang et al., 2024). In this context, a sparse learnable-temperature alternative with $\alpha$-entmax have been proposed as a direct way to mitigate these issues (Vasylenko et al., 2025).

## 6. Conclusion

In this work, we introduced ADASPLASH-2, a hardware-aware efficient implementation of $\alpha$-entmax attention. Our key idea is to construct a compact histogram of streamed attention scores in on-chip SRAM, producing a strong initialization for the $\alpha$-entmax threshold $\tau$, and reducing refinement to a small (typically 1-2) number of steps while integrating naturally with GPU execution. To leverage sparsity, ADASPLASH-2 uses a lightweight bitpacked block mask to skip zero blocks efficiently. Empirically, ADASPLASH-2 delivers training speedups over prior $\alpha$-entmax kernels and, in moderate-to-high block-sparsity regimes, matches or exceeds FlashAttention-2, with gains driven primarily by a substantially faster backward pass. In language modeling, $\alpha$-entmax models match or improve short-context performance and deliver clear gains on long-context evaluations.

## Impact Statement

Our method, ADASPLASH-2, provides an efficient implementation of $\alpha$-entmax attention. Efficient attention mechanisms are crucial for scaling transformers to handle long-context sequences. As a result, the improved efficiency has potential applications in large-scale NLP applications, especially in cases where sparsity can be leveraged to reduce computational costs such as in long-context modeling. We do not foresee direct societal consequences from our method itself, but its integration into decision-making models may still reflect biases in training data. As such, we encourage careful evaluation when deploying models trained with ADASPLASH-2 in high-stakes applications, ensuring that efficiency gains do not overcome ethical concerns.

## Acknowledgments

We would like to the SARDINE lab team for the helpful discussions. This work was supported by the project DECOLLAGE (ERC-2022-CoG 101088763), by the Portuguese Recovery and Resilience Plan through project C645008882-00000055 (Center for Responsible AI), and by FCT/MECI through national funds and when applicable co-funded EU funds under UID/50008/2025 – Instituto de Telecomunicações (DOI 10.54499/UID/50008/2025). We thank EuroHPC and FCT/FCCN for the HPC resources used to support this work through grant EuroHPC Challenge AI-Boost Project TowerMoE in the Leonardo supercomputer at CINECA and POR2025.00272.CPCA.A3 in Mare Nostrum 5 at BSC, respectively. Vlad Niculae is supported by the Dutch Research Council (NWO) via VI.Veni.212.228. Edoardo M. Ponti is supported by the ERC Starting Grant AToM-FM (101222956).

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

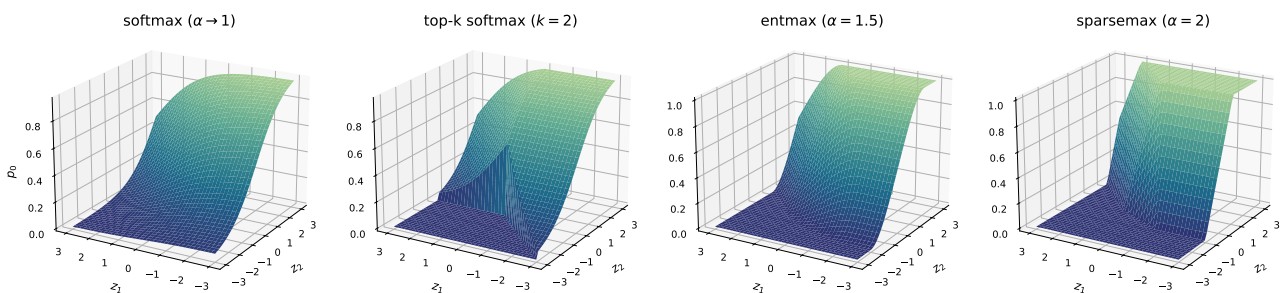

*Figure 5.* Visualization of $\alpha$-entmax for different values of $\alpha$. We also include top-$k$ softmax with $k = 2$ for completeness. Each panel shows how the probability mass of $p_0$ varies for the input $\boldsymbol{z} = [0, z_1, z_2]$. For softmax, $p_0$ is always non-zero, regardless of $z_1$ and $z_2$. As $\alpha$ increases, $\alpha$-entmax increasingly assigns exactly zero probability to $z_0$. While $\alpha$-entmax changes smoothly with the scores (yielding piecewise-smooth gradients), top-$k$ softmax changes the "selected set" abruptly and, consequently, these non-differentiable boundaries induce discontinuous gradients that might lead to training instabilities.

## A. $\alpha$-entmax Transformation

The $\alpha$-entmax transformation of a score vector $\boldsymbol{z} \in \mathbb{R}^n$ is defined as follows (Peters et al., 2019):

$$\alpha\text{-entmax}(\boldsymbol{z}) := \arg \max_{\boldsymbol{p} \in \triangle_n} \boldsymbol{p}^\top \boldsymbol{z} + H_\alpha(\boldsymbol{p}), \quad \triangle_n := \{\boldsymbol{p} \in \mathbb{R}^n : \boldsymbol{p} \geq \boldsymbol{0}, \boldsymbol{1}^\top \boldsymbol{p} = 1\}, \tag{12}$$

where $H_\alpha(\boldsymbol{p})$ is the Tsallis($\alpha$) entropy (Tsallis, 1988). Solving the optimization problem above corresponds to finding a threshold $\tau$ so that $\boldsymbol{p}^\star$ sums to 1. Given $\tau$, we can easily evaluate $\alpha$-entmax as per Equation 3, which we re-state here for easiness:

$$\alpha\text{-entmax}(\boldsymbol{z}) = [(\alpha - 1)\boldsymbol{z} - \tau\boldsymbol{1}]_+^{1/\alpha-1}, \tag{13}$$

where $[\cdot]_+$ is the ReLU function. Figure 5 illustrates how $\alpha$-entmax($\boldsymbol{z}$) behaves for different choices of $\alpha$. From the equation above, it is clear that coordinates with $(\alpha - 1)s_i \leq \tau$ become exactly zero. In other words, $\alpha$-entmax yields dynamic sparsity, where the pattern of zeros depends on the input. However, this flexibility comes at a computational cost: unlike softmax, $\alpha$-entmax does not have a direct closed-form solution but rather requires more involved methods, which we discuss next.

$\alpha$-**entmax computation.** Order-based algorithms have only been proposed for $\alpha = 2$ (Michelot, 1986; Duchi et al., 2008; Condat, 2016) and $\alpha = 1.5$ (Peters et al., 2019). This family of algorithms builds upon the equivalence between $\tau$ and the "active set", denoted as $\mathcal{S} = \{j : z_j > \tau\}$. At $\tau^\star$, this set indexes the nonzero terms in the expression of $f$, or, equivalently, the nonzero elements in the $\alpha$-entmax output. If the active set contains an index, it must contain all indices with greater or equal value, so there are $n$ possible active sets $\mathcal{S}_1, \ldots, \mathcal{S}_n$, where contains the indices of the $k$ largest values of $\boldsymbol{z}$. Moreover, since the output must be a valid probability distribution, $\mathcal{S}_k$ is never empty. At the true support size $k^\star$, we have

$$f(\tau) = f_{k^\star}(\tau) := -1 + \sum_{j \in \mathcal{S}_{k^\star}} (z_j - \tau)^{1/\alpha-1}. \tag{14}$$

Note that the ReLU function is no longer necessary as all terms are strictly positive. Finding the root of $f_k$ for any $k$ can be done efficiently in the case of $\alpha = 2$ (linear equation) and $\alpha = 1.5$ (quadratic equation).

In contrast, root-finding algorithms apply for all values of $\alpha$. Since $\max(\boldsymbol{z}) = 1$, following Peters et al. (2019), we have

$$0 \leq \tau^\star \leq 1 - n^{1-\alpha}. \tag{15}$$

Moreover, since $f$ is continuous and $f(0) \leq 0$ and $f(1 - n^{1-\alpha}) \geq 0$, the root must be found in the interval. Blondel et al. (2019) propose a bisection or binary search approach to finding $\tau^\star$. Similar in spirit, Gonçalves et al. (2025) introduces a GPU-oriented solver of $\alpha$-entmax that uses a hybrid *Halley-bisection* method, which combines the fast local convergence of higher-order root-finding (Scavo & Thoo, 1995) with the convergence guarantees of bisection.

# B. Proofs

Throughout, we let $\alpha > 1$, represent scores as $s \in \mathbb{R}^n$, and we work with centered and scaled scores $z = (\alpha-1)s - (m-1)\mathbf{1}$, where $m = (\alpha - 1)\max(s)$, which ensures $\max(z) = 1$. Also, we recall from Equation 15 that $\tau \geq \max(z) - 1 = 0$ and $\tau \leq \max(z) - n^{1-\alpha} = 1 - n^{1-\alpha}$.

## B.1. Proof of Proposition 1

*Proof of Proposition 1.* We divide the proof in four parts.

**Step 1: Establishing the sandwich inequality.** By construction of the binning scheme, for any score $z_j \in [0, 1]$ assigned to bin $k$, we have:

$$\frac{k}{B} \leq z_j < \frac{k+1}{B}, \tag{16}$$

which implies:

$$b_j \leq z_j < b_j + h, \quad \text{where } h = \frac{1}{B}, \tag{17}$$

where $b_j := \frac{k}{B}$ denotes the left edge of the bin to which $z_j$ is assigned.

For scores $z_j < 0$, we have $[z_j - \tau]_+ = 0 = [b_j - \tau]_+$ for all $\tau > 0$, so these scores do not affect the proof. Consider the function $\phi(t) = [t]_+^{\frac{1}{\alpha-1}}$ for $t \in \mathbb{R}$, which is monotone non-decreasing and continuous for $\alpha > 1$. For any fixed $\tau \in [0, 1]$, using the monotonicity of $\phi$ and inequality in Eq. 17:

$$b_j - \tau \leq z_j - \tau < b_j + h - \tau \tag{18}$$

$$\implies [b_j - \tau]_+ \leq [z_j - \tau]_+ < [b_j + h - \tau]_+ \tag{19}$$

$$\implies \phi(b_j - \tau) \leq \phi(z_j - \tau) < \phi(b_j + h - \tau). \tag{20}$$

The strict inequality on the right holds because: if $z_j - \tau > 0$, then $z_j < b_j + h$ implies $z_j - \tau < b_j + h - \tau$, and $\phi$ is strictly increasing on $(0, \infty)$.

Summing over all $j = 1, \ldots, n$:

$$\sum_{j=1}^{n} \phi(b_j - \tau) \leq \sum_{j=1}^{n} \phi(z_j - \tau) < \sum_{j=1}^{n} \phi(b_j + h - \tau). \tag{21}$$

Subtracting 1 from each term:

$$f_h(\tau) \leq f(\tau) < f_h(\tau - h), \quad \forall \tau \in [0, 1], \tag{22}$$

where we used the fact that:

$$\sum_{j=1}^{n} \phi(b_j + h - \tau) = \sum_{j=1}^{n} \phi(b_j - (\tau - h)) = f_h(\tau - h) + 1. \tag{23}$$

**Step 2: Properties of $f$ and $f_h$.** Both $f(\tau)$ and $f_h(\tau)$ are:

- Continuous on $[0, 1]$

- Strictly decreasing on $[0, 1]$ (since $\phi$ is strictly increasing where positive)

- Satisfy $f(0) > 0$ and $f(1) < 0$ (similarly for $f_h$)

By the intermediate value theorem, there exist unique roots:

$$\tau^\star : \quad f(\tau^\star) = 0, \tag{24}$$

$$\tau_h : \quad f_h(\tau_h) = 0. \tag{25}$$

**Step 3: Bounding $\tau_h$ relative to $\tau^\star$.** Evaluating the sandwich inequality in Eq. 22 at $\tau = \tau^\star$:

$$f_h(\tau^\star) \leq f(\tau^\star) < f_h(\tau^\star - h). \tag{26}$$

Since $f(\tau^\star) = 0$, we have:

$$f_h(\tau^\star) \leq 0 < f_h(\tau^\star - h). \tag{27}$$

Because $f_h$ is strictly decreasing and continuous, and $f_h(\tau^\star - h) > 0$ while $f_h(\tau^\star) \leq 0$, there must exist a unique $\tau_h \in [\tau^\star - h, \tau^\star]$ such that $f_h(\tau_h) = 0$.

More precisely:

- If $f_h(\tau^\star) < 0$, then $\tau_h \in (\tau^\star - h, \tau^\star)$ by the intermediate value theorem.

- If $f_h(\tau^\star) = 0$, then $\tau_h = \tau^\star$.

In both cases, we have:

$$\tau^\star - h < \tau_h \leq \tau^\star. \tag{28}$$

**Step 4: Error bound.** The absolute error is:

$$|\tau^\star - \tau_h| = \tau^\star - \tau_h \leq h = \frac{1}{B}. \tag{29}$$

This completes the proof. $\qquad\square$

The proof shows that the histogram approximation is *conservative* as it never overestimates $\tau^\star$, which means $\tau_h \leq \tau^\star$. This ensures that the approximate solution produces a sparsity pattern that preserves the support of the real solution—an useful property for safe block-masking.

### B.2. Proof of Proposition 2

We divided the proof of Proposition 2 into two parts: (i) continuous differentiability of the threshold map $f$, and (ii) continuous differentiability of the $\alpha$-entmax transformation (from scores into probabilities). Throughout, we assume $\alpha > 1$, and work with translated and scaled scores $\boldsymbol{z} = (\alpha - 1)\boldsymbol{s} - (m - 1)\mathbf{1}$ where $m = (\alpha - 1)\max(\boldsymbol{s})$. The domain of possible such scores after translation is $D = \{\boldsymbol{z} \in \mathbb{R}^n : \max(\boldsymbol{z}) = 1\}$. Scaling and centering are continuous smooth operations and thus composing with them does not affect the continuity of any order of differentiation. We denote the active set induced by a threshold $\tau$ as $\mathcal{S}_\tau := \{i \in [n] : z_i > \tau\}$.

**Part (i): continuity of $f$.** Given the active set $\mathcal{S}_\tau$, the derivatives of $f$ have the form:

$$f(\tau) = -1 + \sum_{i \in \mathcal{S}_\tau} (z_i - \tau)^{\frac{1}{\alpha - 1}} \tag{30}$$

$$f'(\tau) = -\frac{1}{\alpha - 1} \sum_{i \in \mathcal{S}_\tau} (z_i - \tau)^{\frac{1}{\alpha - 1} - 1} \tag{31}$$

$$f''(\tau) = \frac{1}{\alpha - 1}\left(\frac{1}{\alpha - 1} - 1\right) \sum_{i \in \mathcal{S}_\tau} (z_i - \tau)^{\frac{1}{\alpha - 1} - 2} \tag{32}$$

$$f^{(k)}(\tau) = (-1)^k \left[\prod_{i=0}^{k-1}\left(\frac{1}{\alpha - 1} - i\right)\right] \sum_{i \in \mathcal{S}_\tau} (z_i - \tau)^{\frac{1}{\alpha - 1} - k} \tag{33}$$

We can study the continuity of any such derivative by considering the sum

$$s_k(\tau) := \sum_{i \in \mathcal{S}_\tau} (z_i - \tau)^{\frac{1}{\alpha - 1} - k}, \tag{34}$$

because every term can be written as a continuous transformation of $s_k$. For a small enough $\epsilon > 0$, $S_{\tau+\epsilon} = S_\tau$. As a sum of continuous functions we therefore have

$$\lim_{\epsilon \to 0_+} s_k(\tau + \epsilon) = s_k(\tau). \tag{35}$$

If $\tau \neq z_i$ for any $i \in [n]$ then $S_{\tau-\epsilon} = S_\tau$ also, so the only possible discontinuities are when $\tau = z_i$ for some $i$. In this case, $z_i$ enters the support (and thus the sum) alongside any other tied $z_j = z_i$. Let's say the value $z_i$ appears $n_i \geq 1$ times in the vector. We can write

$$s_k(\tau - \epsilon) = \sum_{j \in S_{\tau-\epsilon}} (z_j - \tau + \epsilon)^{\frac{1}{\alpha-1} - k} \tag{36}$$

$$= \sum_{j \in S_\tau} (z_j - \tau + \epsilon)^{\frac{1}{\alpha-1} - k} + n_i (z_i - \tau + \epsilon)^{\frac{1}{\alpha-1} - k} \tag{37}$$

Taking the limit to 0, the sum is a sum of continuous functions and so we have

$$\lim_{\epsilon \to 0_+} s_k(\tau - \epsilon) = s_k(\tau) + \lim_{\epsilon \to 0_+} n_i(z_i - \tau + \epsilon)^{\frac{1}{\alpha-1} - k} \tag{38}$$

$$= s_k(\tau) + n_i \lim_{\epsilon \to 0_+} \epsilon^{\frac{1}{\alpha-1} - k} \tag{39}$$

It follows that $s_k$ is continuous at $\tau$ iff. $(\alpha - 1)^{-1} - k > 0$. In particular:

- $f(\tau)$ is continuous when $1 < \alpha$.

- $f'(\tau)$ is continuous when $1 < \alpha < 2$.

- $f''(\tau)$ is continuous when $1 < \alpha < 3/2$, and so on.

**Part (ii): continuity of $\alpha$-entmax.** Let

$$F_\alpha(\mathbf{z}, \tau) = -1 + \sum_{i \in S_\tau} (z_i - \tau)^{\frac{1}{\alpha-1}}. \tag{40}$$

Recall that, by continuity and monotonicity of $f$, for any $\mathbf{z}$ there is a unique $\tau \in [0, 1]$ such that $F_\alpha(\mathbf{z}, \tau) = 0$; denote by $g(\mathbf{z})$ the function mapping $\mathbf{z}$ to that corresponding unique root $\tau$.[6] If $\alpha \in (1, 2)$, from the continuity of $f$ shown above, we have that $F_\alpha$ is $\mathcal{C}^1$. The condition that the jacobian be nonsingular in this case is equivalent to $f'(\tau) \neq 0$ (Eq. 31). The terms in the sum $s_k$ are all positive. For $\tau \in [0, 1]$ the sum contains at least one term corresponding to the maximum (scaled) score and thus $f'(\tau) < 0$, confirming nonsingularity. The conditions of the implicit function theorem (Dontchev & Rockafellar, 2014, Thm. 1B.1) are therefore satisfied, which implies the solution mapping $g$ is $\mathcal{C}^1$ for any $\mathbf{z} \in D$. Since $\alpha$-entmax can be written coordinate-wise as

$$p_j(\mathbf{z}) = [z_j - g(\mathbf{z})]_+^{\frac{1}{\alpha-1}}, \tag{41}$$

its (almost everywhere) gradient with respect to $\mathbf{z}$ is

$$\nabla p_j(\mathbf{z}) = \frac{1}{\alpha - 1} [z_j - g(\mathbf{z})]_+^{\frac{1}{\alpha-1} - 1} (\mathbf{e}_j - \nabla g(\mathbf{z})), \tag{42}$$

and $\nabla p_j(\mathbf{z}) = \mathbf{0}$ whenever $z_j < g(\mathbf{z})$. For $1 < \alpha < 2$ this function is a composition/product of continuous functions, and thus it is continuous (Rudin, 1976, Theorems 4.7 & 4.9). Therefore, for this range of $\alpha$, the "almost everywhere" above becomes "everywhere".

Furthermore, by the higher-order extension of the implicit function theorem (Dontchev & Rockafellar, 2014, Prop. 1B.5), we can repeat this argument for higher-order derivatives. When $\alpha < \frac{k+1}{k}$, from the result in the first part we have $F_\alpha \in \mathcal{C}^k$. This implies $g$ is also $\mathcal{C}^k$.

Applying the same argument coordinate-wise, it follows that $\alpha$-entmax is $\mathcal{C}^k$ whenever $1 < \alpha < \frac{k+1}{k}$. Indeed, as $\alpha \to 1_+$ we recover softmax, which is $\mathcal{C}^\infty$.

---

[6]At the root the active set is nonempty: if $S_\tau = \varnothing$, then $F_\alpha(\mathbf{z}, \tau) = -1 \neq 0$. Equivalently, since $\max(\mathbf{z}) = 1$, any root satisfies $\tau < 1$ (because $F_\alpha(\mathbf{z}, \tau) = -1$ for $\tau \geq 1$).

| Bin 0 | Bin 1 | Bin 2 | Bin 3 | Bin 4 | Bin 5 | Bin 6 | Bin 7 |
|-------|-------|-------|-------|-------|-------|-------|-------|
| 0     | 4     | 1     | 2     | 0     | 0     | 1     | 0     |

Bit representation: 0x0001000002010400

*Figure 6.* Example of a bitpacked histogram with $B = 8$ bins and $b = 8$ bits per bin. Each colored segment represents a bin's count encoded in 8 bits of a `uint64` integer.

## C. ADASPLASH-2 Implementation Details

This appendix provides a detailed exposition of ADASPLASH-2's implementation, including the bitpacking schemes that enable efficient histogram construction and block mask traversal. We begin by establishing notation (§C.1), then describe the histogram construction and capacity analysis (§C.2), block mask encoding (§C.4), and overflow handling strategies (§C.5).

### C.1. Notation and Problem Setup

**Sequence and Tiling Parameters.** We work with input sequences of length $n$, head dimension $d$, batch size $B$, and $N_H$ attention heads. The sequences are partitioned into tiles: query tiles of size $B_r$ and key/value tiles of size $B_c$, yielding $T_r = \lceil n/B_r \rceil$ query tiles and $T_c = \lceil n/B_c \rceil$ key/value tiles. Throughout, we use indices $i \in [T_r]$ for query tiles and $j \in [T_c]$ for key tiles.

**Histogram Parameters.** The histogram has shape $\mathcal{H} \in \mathbb{N}^{B_r \times B}$. To avoid atomics or reductions, we first accumulate the counts in local histograms $\mathcal{H}^{\text{local}}$ private to each position, providing $B_r \times B_c$ parallel accumulators. Our local histograms use $B$ bins encoded in a $w$-bit unsigned integer (e.g., $w = 64$ for `uint64`), allocating $b = w/B$ bits per bin. The bin width (resolution) is $h = 1/B$.

**Bitwise Operations.** We use standard notation for bit manipulation:

- $x \ll k$: Left shift by $k$ bits (multiply by $2^k$)

- $x \gg k$: Right shift by $k$ bits (divide by $2^k$, rounded down)

- $x \wedge y$: Bitwise AND

- $x \vee y$: Bitwise OR

- `popc`$(x)$: Population count (number of 1-bits)

- `fns`$(x, k)$: Find first set bit at position $\geq k$

### C.2. Histogram Construction and Bitpacking

The histogram construction in Algorithm 1 (Phase 2) is the computational core of ADASPLASH-2's threshold approximation. We now describe its implementation in detail.

**Bitpacked Encoding Scheme.** Each local histogram accumulator $\mathcal{H}^{\text{local}}_{ik} \in \{0, 1, \ldots, 2^w - 1\}$ encodes $B$ bin counts by partitioning its $w$ bits into $B$ equal segments of $b = w/B$ bits each. The value $\mathcal{H}^{\text{local}}_{ij}$ can be decomposed as:

$$\mathcal{H}^{\text{local}}_{ij} = \sum_{t=0}^{B-1} c_k \, 2^{kb}, \quad \text{where } c_k \in [0, 2^b - 1] \text{ is the local counts for bin } t. \tag{43}$$

Figure 6 illustrates this structure for $B = 8$ bins with $b = 8$ bits per bin in a `uint64`.

**Accumulation.** For each entry $Z_{ij}$ (see Eq. 6), we compute the corresponding bin,

$$k_{ij} \;=\; \min\big(\max(\lfloor B\,Z_{ij}\rfloor, 0),\, B-1\big). \tag{44}$$

Each update increments the respective bit field inside the packed integer:

$$\mathcal{H}_{ij}^{\text{local}} \;\leftarrow\; \mathcal{H}_{ij}^{\text{local}} \;+\; \mathbf{1}[Z_{ij} \geq 0] \cdot \big(1 \ll (b\,k_{i,j})\big). \tag{45}$$

In Triton, this update can be achieved with a shift plus an integer addition.

**Extraction and Aggregation.** After accumulation, we extract the count for bin $k \in \{0, \ldots, B-1\}$ from the packed `uint64` word by shifting and masking. Let $\mathcal{B}_b = 2^b - 1$ be the $b$-bit mask (i.e., $b$ consecutive 1-bits). Then, we can aggregate across the $B_c$ parallel accumulators (over $j$) to obtain per-query bin counts

$$\mathcal{H}_{ik} \;=\; \sum_j \big(\mathcal{H}_{ij}^{\text{local}} \gg (k\,b)\big) \;\wedge\; \mathcal{B}_b. \tag{46}$$

In Triton, we vectorize the shift-and-mask over $b$ using a compile-time range.

### C.3. Histogram initialization

With the final histogram $\mathcal{H}$ obtained, we can solve the approximate problem (see Eq. 9) using a modified version of the sorting-based algorithms discussed in Section 2.4. The $\tau_0$ and bracket $[\tau_{\text{lo}}, \tau_{\text{hi}}]$ obtained from this phase will be used for posterior refinement. Here, we present the algorithms in detail.

**Case $\alpha \in \{1.5, 2.0\}$.** As the counts are conveniently sorted in the histogram $\mathcal{H}$, we scan bins in descending order, $k \in [B-1, .., 0]$. We maintain prefix sums over the active (ordered) set $\mathcal{P} = [B-1, B-2, \ldots, k]$:

$$S_0 = \sum_{k \in \mathcal{P}} \mathcal{H}_k, \quad S_1 = \sum_{k \in \mathcal{P}} \mathcal{H}_k \cdot \frac{k}{B}, \quad S_2 = \sum_{k \in \mathcal{P}} \mathcal{H}_k \cdot \left(\frac{k}{B}\right)^2. \tag{47}$$

Intuitively, we start with an empty set where $f_h(\tau) = -1$. Iteratively, we add contributions from the next bin ($k = B-1$, then $k = B-2$, ...) while evaluating $f(\tau = k/B)$. If adding the contribution of a bin, $k$, cause $f(\tau)$ to be positive, it means that the root is between the values $k/B < \tau < (k+1)/B$. When that set $\mathcal{P}_k^\star$ is found, we can drop the ReLU, $[\cdot]_+$, and solve for $\tau$. Example for $\alpha = 2.0$:

$$\sum_{k \in \mathcal{P}_k^\star} \mathcal{H}_k \left(\frac{k}{B} - \tau\right) = 1$$

$$\underbrace{\sum_{k \in \mathcal{P}_k^\star} \mathcal{H}_k \frac{k}{B}}_{S_1} - \tau \underbrace{\sum_{k \in \mathcal{P}_k^\star} \mathcal{H}_k}_{S_0} = 1 \implies \tau = \frac{S_1 - 1}{S_0}$$

**General $\alpha$.** For a general $\alpha$, we can find $\tau^\star$ by using any method originally developed for $\alpha$-entmax. However, instead of iterating over the actual scores, we can cheaply iterate over the histogram $\mathcal{H}$.

### C.4. Block Mask Encoding and Traversal

After computing the threshold $\tau$ in Phase 3 of Algorithm 1, we construct a binary mask $\mathcal{M} \in \{0,1\}^{T_r \times T_c}$ indicating which tile pairs $(i, j)$ contain non-zero attention weights. Naively storing this as a boolean tensor would require $T_r \times T_c$ bytes. Instead, we use bitpacking encoded in `int32` to reduce memory by a factor of $32\times$. That is, we pack 32 consecutive column indices into a single `int32`, yielding a compact representation $\mathcal{M}^{\text{packed}} \in \mathbb{Z}^{T_r \times \lceil T_c/32 \rceil}$ defined by:

$$\mathcal{M}_{i, \lfloor j/32 \rfloor}^{\text{packed}} = \sum_{k=0}^{31} \mathcal{M}_{i, 32\lfloor j/32 \rfloor + k} \cdot 2^k. \tag{48}$$

As a result, each bit in $\mathcal{M}_{i,m}^{\text{packed}}$ indicates whether the corresponding tile pair has non-zero weights.

*Table 4.* Histogram capacity analysis for different integer types and bin configurations with $B_c = 64$. Capacity indicates maximum keys per query tile; resolution $h = 1/B$ is the bin width affecting threshold accuracy (Proposition 1).

| Type ($w$) | Bins ($B$) | Bits/Bin ($b$) | Max/Bin | Capacity | Resolution ($h$) |
|---|---|---|---|---|---|
| *uint64* | | | | | |
| 64 | 4 | 16 | 65,535 | 4,194,240 | 0.25 |
| 64 | 8 | 8 | 255 | 16,320 | 0.125 |
| 64 | 16 | 4 | 15 | 960 | 0.0625 |
| *uint128* | | | | | |
| 128 | 8 | 16 | 65,535 | 4,194,240 | 0.125 |
| 128 | 16 | 8 | 255 | 16,320 | 0.0625 |
| 128 | 32 | 4 | 15 | 960 | 0.03125 |

**Efficient Traversal with GPU Instructions.** To iterate through active tiles, we use two GPU-native instructions:

1. popc($\mathcal{M}_{i,m}^{\text{packed}}$): Returns the number of set bits (active tiles) in the int32.

2. fns($\mathcal{M}_{i,m}^{\text{packed}}, k$): Find the n-th set bit given by offset $k$.

The traversal procedure processes only the $|\mathcal{M}|$ non-zero tiles rather than all $T_r \times T_c$ potential tiles. Concretely, for each packed word index $m$, we iterate only over set bits (active key blocks) using popc to count and fns to locate bits. If bit_pos is the position of a set bit within word $m$, the corresponding key-block index is

$$j = 32m + \text{bit\_pos}. \tag{49}$$

The operations operations are implemented via Triton's inline PTX assembly, making the traversal cost negligible compared to the memory and compute operations. This traversal mechanism is used in Phase 4 of Algorithm 1 to load only the non-zero key and value tiles, and similarly in the backward pass to skip zero blocks.

### C.5. Histogram Capacity and Overflow Handling

The bitpacking scheme naturally imposes capacity limits for our histogram since each bin can count up to $2^b - 1$ items before overflow. With $B_c$ parallel accumulators per query tile, the maximum capacity is:

$$C_{\max} = B_c \times (2^{w/B} - 1). \tag{50}$$

Table 4 summarizes capacity and resolution trade-offs for various configurations. The table reveals a fundamental trade-off: increasing $B$ improves threshold resolution (tighter bound in Proposition 1) but reduces capacity per bin. For sequences up to 16K tokens per query, uint64 with $B = 8$ provides excellent resolution ($h = 0.125$) while avoiding overflow. For longer sequences, we turn to an overflow mitigation strategy, which we describe next.[7]

**Periodic Flush.** After processing $C_{\max}$ keys, we extract all bin counts, accumulate them into shared memory array of size $(T_r \times B)$, and reset the SRAM histogram to zero. Since each query processes $T_c$ key tiles and the histogram saturates after $2^b - 1$ tiles, the number of flushes required for a sequence of length $n$ is:

$$N_{\text{flush}} = \left\lceil \frac{n}{B_c \times (2^b - 1)} \right\rceil = \left\lceil \frac{T_c}{2^b - 1} \right\rceil. \tag{51}$$

### C.6. Backward Pass Algorithms

In the backward pass, we exploit the block mask $\mathcal{M}$ to skip zero gradients, achieving asymptotic speedups when sparsity is high. The key difference from standard attention backpropagation is the use of the block mask $\mathcal{M}$ (and its transpose $\mathcal{M}^\top$) to iterate only over non-zero blocks. For $|\mathcal{M}| \ll T_r \cdot T_c$ (high sparsity), this provides order-of-magnitude speedups in the backward pass, offsetting the higher forward pass cost of ADASPLASH-2 relative to FlashAttention-2.

---

[7]We note that uint128 is already available in CUDA: https://developer.nvidia.com/blog/implementing-high-precision-decimal-arithmetic-with-cuda-int128/

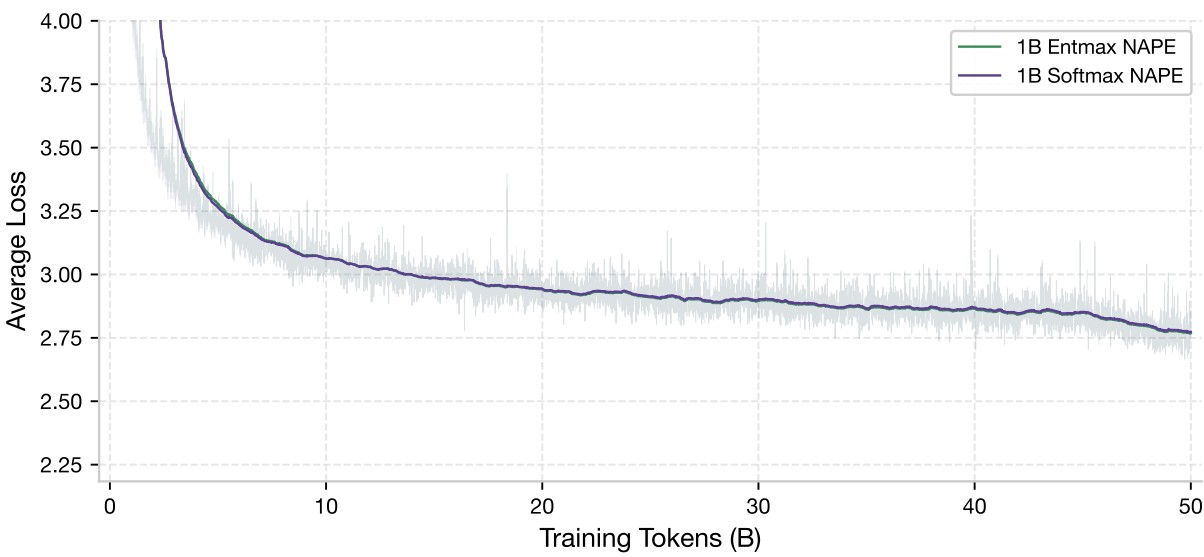

*Figure 7.* Loss curves recorded during the language modeling training runs. Entmax NAPE shows consistently lower loss towards the later stages of training. Both models are generally stable as no harsh loss spikes occurred.

## D. Language Modeling

We train our language models with the *torchtitan* library (Liang et al., 2025) and in a single node with 4x H100 NVIDIA GPUs. The models are trained for 50B tokens of DCLM-Edu data with the WSD scheduler (2000 warmup steps, stable until 80% of training and decay for the latter 10B tokens). Weight decay is set to 0.1 and maximum learning rate is set to $3 \times 10^{-4}$ and $2 \times 10^{-4}$ for 350M and 1B parameter models, respectively. Final learning rate is set to 10% of the maximum. Context length is set to 4096 tokens with an effective batch size of 524k tokens, resulting in 100k total steps. We decide to go beyond the Chinchilla (Hoffmann et al., 2022) optimal token counts since we saw steady increased model performance even with the additional tokens. Training runs were stable throughout, with both softmax-based baseline and ADASPLASH-2 models registering no noticeable loss spikes, as can be seen in Figure 7. Finally, for inference we use a simplified variant of ADASPLASH-2 that operates on fully materialized logits $z \in \mathbb{R}^n$. Despite its simplicity, in practice we found this approach to be faster than parallelizing over the K/V sequence dimension for the sequence lengths considered in this work.

**Context-length extension.** For the long-context extension phase we swap the data mixture at the decay phase (10B tokens) with the ProLong dataset (Gao et al., 2025), keeping total training data at 50B tokens. Here, we increase the sequence length to 32k and keep the effective batch size fixed. For RoPE-based models, we increase their $\theta$ from the 50k default to 800k, consistent with the ProLong methodology.

**Evaluation.** For long-context we adopt both the full RULER benchmark (Hsieh et al., 2024) and the In-Context Learning subset of HELMET (Yen et al., 2025) which is a capability not covered by RULER. For evaluation in short-context tasks, we also include Lambada (Paperno et al., 2016; Radford et al., 2019) for evaluating perplexity, and drop BoolQ (Clark et al., 2019) from the evaluation mix following Olmo3 (Olmo et al., 2025). For completeness, we provide the results of all of our models on short-context benchmarks in Table 5. The results on RULER for the long-context adapted models are presented in Table 1, while the HELMET-ICL results can be seen in Table 2.

**Discussion.** Overall, we see that extended models show a small degradation on short-context benchmarks, which is typical of long-context extension (Gao et al., 2025). However, the scenario flips when we look at Table 1, where we observe that our long-context procedure was effective and 32K-context models are able to complete tasks at 32K context length to an effective degree. Analysis of In-Context Learning capabilities (Table 2) between dense and sparse models further confirm the strengths of ADASPLASH-2 as it outperforms softmax models across all evaluated sequence lengths.

*Table 5.* Downstream evaluations on short-context benchmarks. Models are trained with 50B tokens from DCLM-Edu and have 4K context length. Those marked with *(32K)* undergo context-length extension with 10B tokens of Prolong data. Best results are in **bold**.

| Model | LMB (ppl) | LMB | ARC-E | ARC-C | CSQA | HS | OBQA | PIQA | SocialQA | WG | Avg. |
|---|---|---|---|---|---|---|---|---|---|---|---|
| *350M params.* | | | | | | | | | | | |
| Softmax with RoPE | 23.93 | 40.6 | 61.9 | 32.1 | 50.5 | 41.6 | 38.4 | 66.6 | 45.1 | 48.9 | 47.3 |
| Entmax with RoPE | 22.36 | 39.5 | 63.0 | 34.8 | 50.5 | 40.2 | 39.0 | 64.8 | 44.1 | 53.4 | 47.7 |
| Softmax with NAPE | 19.23 | 41.2 | 61.8 | 33.4 | 47.6 | 40.5 | 38.0 | 64.8 | 43.5 | 52.7 | 47.1 |
| Entmax with NAPE | 18.62 | 42.4 | 61.9 | 33.0 | 51.2 | 41.0 | 39.2 | 66.0 | 44.4 | 53.7 | **48.1** |
| *1B params.* | | | | | | | | | | | |
| Softmax with RoPE | 15.01 | 44.7 | 69.0 | 36.0 | 58.7 | 49.6 | 42.8 | 70.0 | 46.4 | 55.6 | 52.5 |
| Entmax with RoPE | 15.76 | 43.9 | 65.4 | 36.6 | 56.2 | 47.4 | 41.6 | 69.8 | 45.7 | 54.2 | 51.2 |
| Softmax with NAPE | 11.97 | 48.0 | 69.3 | 37.4 | 56.7 | 48.8 | 45.2 | 69.5 | 46.5 | 55.2 | 53.0 |
| Entmax with NAPE | 11.42 | 49.2 | 67.7 | 39.9 | 57.3 | 48.7 | 45.0 | 68.3 | 47.1 | 55.1 | **53.1** |
| *1B params.* | | | | | | | | | | | |
| Softmax with RoPE (32K) | 14.95 | 46.8 | 61.9 | 34.0 | 50.0 | 44.4 | 42.0 | 68.0 | 45.1 | 53.2 | 49.5 |
| Entmax with RoPE (32K) | 12.82 | 48.2 | 63.7 | 34.1 | 50.2 | 45.0 | 40.4 | 67.1 | 45.8 | 52.5 | 49.7 |
| Softmax with NAPE (32K) | 12.04 | 50.7 | 62.4 | 33.6 | 49.8 | 45.4 | 41.0 | 68.2 | 46.5 | 51.9 | 49.9 |
| Entmax with NAPE (32K) | 13.01 | 49.0 | 64.1 | 34.2 | 49.2 | 45.8 | 41.8 | 67.6 | 46.9 | 52.9 | **50.2** |

## D.1. Dense to Sparse Attention Conversion

Having established the efficacy and efficiency of sparse attention, we make a preliminary study of how to convert a pretrained softmax model into a $\alpha$-entmax one. We do so via continued pretraining: starting from a 1B-parameter checkpoint trained with softmax + NAPE at 4K context length, we replace softmax with $\alpha$-entmax (implemented with ADASPLASH-2) and continue training for 10B tokens on Nemotron-CC-V2 (Basant et al., 2025), keeping the final learning rate unchanged. We focus only on 1B NAPE model variants due to the seemingly sub-optimality of RoPE for $\alpha$-entmax attention. Table 6 shows the full results. Directly replacing softmax with $\alpha$-entmax in the attention modules without any additional training results in a performance degradation but not a total collapse (see "Entmax (converted)" entry). After only 2B tokens, at 52B tokens in total, the converted checkpoint already outperforms the baseline softmax scores on average. Finally, we continue training the baselines to measure the total performance lost due to the conversion, evaluating after successive increases of 1B tokens. We observe that the finals results are roughly on par with the results obtained by softmax and $\alpha$-entmax models trained from scratch on the full 60B tokens. Overall, this strategy offers insight into a efficient recipe for building entmax-based models from existing softmax + NAPE checkpoints, which is particularly appealing for midtraining or long-context extension phases. Applying this methodology to other positional encodings or architectures could, additionally, require other adaptation phases (Gelberg et al., 2026).

## E. Attention Sparsity and Efficiency

**Sparse attention patterns.** In Figure 8, we analyze emerging attention block sparsity in the 1B Entmax (NAPE) model trained up to 32K context length. We compute the average $64 \times 64$ attention block sparsity ratios across 64 sequences sampled from the ProLong dataset, evaluated at multiple context lengths (4K, 8K, 16K, 32K, 64K and 128K). Across all context lengths, the sparsity pattern is strongly head-dependent. The first half of the heads (0-11; ALiBi) are consistently sparse across layers. In contrast, the second half (12-23; NoPE) is comparatively dense at 4K, but develops *structured* sparsity as context grows, most notably in mid-to-late layers. The overall mean sparsity increases monotonically with context length, indicating that longer contexts are handled with less dense attention on average.

**Context scaling experiment details.** We study the runtime behavior of ADASPLASH-2 as context length increases using attention block sparsity patterns extracted from the 1B Entmax + NAPE model with 32k context length, reported in Table 1. To this end, we select a fixed transformer layer that exhibits high block sparsity and record its average block sparsity as a function of context length. For each context length, we benchmark ADASPLASH-2 at the corresponding sparsity level, run forward and backward passes, and report average runtimes over repeated runs. Figure 4 reports the resulting average forward and backward runtimes as a function of context length.

*Table 6.* Downstream evaluations on short-context benchmarks Models are trained with 50B tokens from DCLM-Edu and have 4K context length. *Entmax (converted)* implies the softmax-based model was evaluated with ADASPLASH-2 without further training, while *Entmax (CPT)* is the converted model subjected to Continuous Pre-Training (CPT).

| Model (*1B params.*) | Tokens | LMB | ARC-E | ARC-C | CSQA | HS | OBQA | PIQA | SocialQA | WG | Avg. |
|---|---|---|---|---|---|---|---|---|---|---|---|
| *50B token baselines* | | | | | | | | | | | |
| Softmax (scratch) | 50B | 48.0 | 69.3 | 37.4 | 56.7 | 48.8 | 45.2 | 69.5 | 46.5 | 55.2 | 53.0 |
| Entmax (scratch) | 50B | 49.2 | 67.7 | 39.9 | 57.3 | 48.7 | 45.0 | 68.3 | 47.1 | 55.1 | 53.1 |
| *softmax → 1.5-entmax* | | | | | | | | | | | |
| Entmax (converted) | 50B | 31.3 | 42.3 | 31.8 | 47.3 | 36.6 | 36.0 | 63.0 | 41.0 | 50.8 | 42.2 |
| Entmax (CPT) | 51B | 46.6 | 69.2 | 37.5 | 56.8 | 48.3 | 47.2 | 69.7 | 47.4 | 53.5 | 52.9 |
| Entmax (CPT) | 52B | 46.9 | 68.8 | 37.7 | 57.0 | 49.2 | 45.8 | 69.8 | 46.7 | 55.2 | 53.0 |
| Entmax (CPT) | 53B | 46.3 | 69.7 | 37.6 | 57.6 | 48.8 | 45.4 | 70.2 | 46.6 | 55.2 | 53.0 |
| Entmax (CPT) | 54B | 46.4 | 69.7 | 38.8 | 56.7 | 50.6 | 46.0 | 70.3 | 46.9 | 54.7 | 53.3 |
| Entmax (CPT) | 55B | 47.5 | 69.6 | 40.0 | 57.6 | 50.6 | 45.8 | 69.9 | 46.8 | 55.6 | 53.7 |
| Entmax (CPT) | 56B | 48.1 | 68.9 | 39.7 | 56.8 | 50.7 | 46.2 | 69.8 | 47.9 | 55.2 | 53.7 |
| Entmax (CPT) | 57B | 47.3 | 69.0 | 39.3 | 57.4 | 50.3 | 46.8 | 70.0 | 46.6 | 55.4 | 53.6 |
| Entmax (CPT) | 58B | 47.6 | 69.1 | 39.6 | 57.8 | 49.8 | 46.4 | 70.6 | 46.9 | 55.1 | 53.7 |
| Entmax (CPT) | 59B | 48.0 | 69.7 | 38.9 | 58.3 | 50.3 | 47.6 | 69.6 | 46.5 | 54.1 | 53.7 |
| Entmax (CPT) | 60B | 47.4 | 68.8 | 39.7 | 57.7 | 50.6 | 45.4 | 69.6 | 47.1 | 55.0 | 53.5 |
| *60B token baselines* | | | | | | | | | | | |
| Softmax (scratch) | 60B | 48.5 | 68.9 | 37.1 | 58.4 | 51.2 | 46.6 | 70.0 | 47.4 | 55.2 | 53.7 |
| Entmax (scratch) | 60B | 47.8 | 69.5 | 41.6 | 58.8 | 50.7 | 47.2 | 68.7 | 48.7 | 57.0 | 54.4 |

*Table 7.* Relative runtime (forward + backward) of ADASPLASH-2 against FlashAttention-2 on a single H100 GPU, for causal attention with increasing $64 \times 64$ block sparsity. Values are normalized to FA2 (Triton); higher is faster.

| | Block sparsity | | | | | |
|---|---|---|---|---|---|---|
| **Method** | **0%** | **60%** | **70%** | **80%** | **90%** | **98%** |
| FA2 (Triton) | **1.00×** | **1.00×** | 1.00× | 1.00× | 1.00× | 1.00× |
| FA2 (CUDA) | 0.74× | 0.74× | 0.74× | 0.74× | 0.74× | 0.74× |
| ADASPLASH-2 | 0.56× | 0.94× | **1.05×** | **1.20×** | **1.40×** | **1.59×** |

**Microbenchmarks on Hopper GPUs.** To verify that the efficiency gains observed on Ampere transfer to the Hopper hardware used in our language-model experiments (§4.3), we additionally benchmark ADASPLASH-2 against FlashAttention-2 on a single H100 GPU using the same increasing-sparsity setup as Figure 1. On H100, the Triton implementation of FlashAttention-2 is the strongest available FA2 baseline, so we use it as the reference. As reported in Table 7, despite ADASPLASH-2 not incorporating any Hopper-specific optimization (e.g., TMA, WGMMA, warp specialization), the same pattern observed on Ampere holds: ADASPLASH-2 is slower in the dense regime, but as block sparsity increases the gap closes and ADASPLASH-2 surpasses FA2 around 65% block sparsity, reaching a 1.59× speedup at 98%. This shows that the algorithmic gains of ADASPLASH-2 carry over to the Hopper hardware used in our language-model experiments.

# F. Full Algorithms

We provide the full forward pass pseudo-code of ADASPLASH-2 in Algorithm 1, and pseudo-code for our two backward kernels in Algorithm 2 and 3.

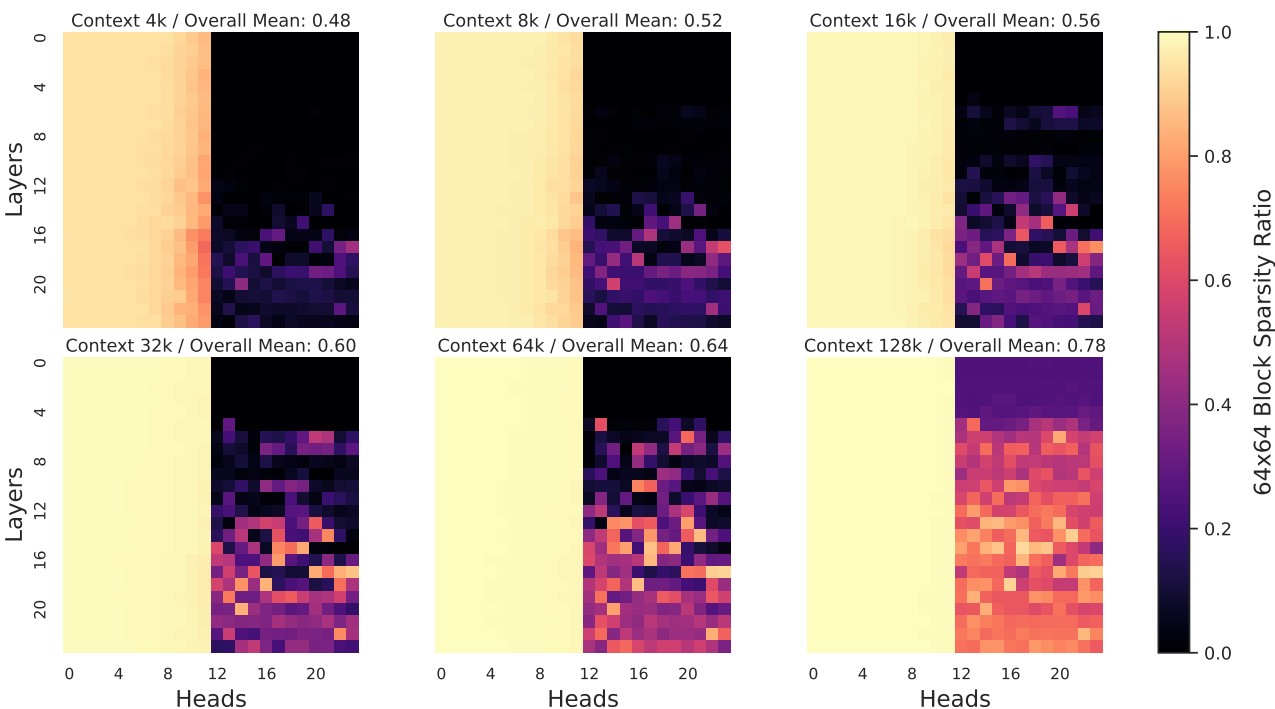

*Figure 8.* Average $64 \times 64$ attention block sparsity ratio for the Entmax (NAPE) model with 32K context length. Panels correspond to evaluated context lengths (4K / 8K / 16K / 32K / 64K / 128K). We report the overall average sparsity across all layers and heads in the title on each plot.

---

**Algorithm 1** ADASPLASH-2 Forward Pass

---

**Require:** Matrices $Q, K, V \in \mathbb{R}^{n \times d}$ in HBM, parameter $\alpha > 1$, bin count $B$, block sizes $(B_r, B_c)$.
**Ensure:** Output $O \in \mathbb{R}^{n \times d}$ in HBM, threshold vector $\tau \in \mathbb{R}^n$ in HBM, block mask $\mathcal{M} \in \mathbb{Z}^{\lceil n/B_r \rceil \times \lceil n/B_c \rceil}$ in HBM.

1: Let $T_r = \lceil n/B_r \rceil$ and $T_c = \lceil n/B_c \rceil$.
2: Initialize $O \leftarrow 0 \in \mathbb{R}^{n \times d}$, $\tau \leftarrow 0 \in \mathbb{R}^n$, and $\mathcal{M} \leftarrow 0 \in \mathbb{Z}^{T_r \times T_c}$ in HBM.
3: Divide $Q, O$ into $T_r$ blocks $Q_i, O_i \in \mathbb{R}^{B_r \times d}$, and divide $K, V$ into $T_c$ blocks $K_j, V_j \in \mathbb{R}^{B_c \times d}$. Divide $\tau$ into $T_r$ blocks $\tau_i \in \mathbb{R}^{B_r}$.
4: **for** $i = 1$ to $T_r$ **do**
5:     Load $Q_i$ from HBM to on-chip SRAM.
6:     **// Phase 1: Compute row-wise maximum**
7:     Initialize $m_i \leftarrow -\infty \in \mathbb{R}^{B_r}$ on SRAM.
8:     **for** $j = 1$ to $T_c$ **do**
9:         Load $K_j$ from HBM to SRAM.
10:         Compute $S_i^{(j)} \leftarrow Q_i K_j^\top \in \mathbb{R}^{B_r \times B_c}$.
11:         Update $m_i \leftarrow \max\big(m_i, \; \max_{\text{col-wise}}(S_i^{(j)})\big)$.
12:     **end for**
13:
14:     **// Phase 2: Build histogram in a single pass**
15:     Initialize local histogram $\mathcal{H}_{ij}^{\text{local}} \leftarrow 0 \in \mathbb{R}^{B_r}$ on SRAM.
16:     **for** $j = 1$ to $T_c$ **do**
17:         Load $K_j$ from HBM to SRAM.
18:         Compute $S_i^{(j)} \leftarrow Q_i K_j^\top \in \mathbb{R}^{B_r \times B_c}$.
19:         Update $\mathcal{H}_{ij}^{\text{local}}$ with quantized scores from $Z_i^{(j)}$.
20:     **end for**
21:     Reduce $\mathcal{H}_{ij}^{\text{local}}$ into $\mathcal{H}_i \in \mathbb{R}^{B_r \times B}$.
22:     $\tau_i \leftarrow \text{SolveHistogram}(\mathcal{H}_i, \alpha) \in \mathbb{R}^{B_r}$         ▷ See §C.3
23:
24:     **// Phase 3: Single hybrid step and save block mask**
25:     Initialize accumulators for $f(\tau_{0,i}), f'(\tau_{0,i}), f''(\tau_{0,i})$ (row-wise).
26:     **for** $j = 1$ to $T_c$ **do**
27:         Load $K_j$ from HBM to SRAM.
28:         Compute $S_i^{(j)} \leftarrow Q_i K_j^\top \in \mathbb{R}^{B_r \times B_c}$.
29:         Compute $P_i^{(j)} \leftarrow \max\big(0, \; (\alpha-1)S_i^{(j)} - \tau_{0,i}\big)^{1/\alpha-1} \in \mathbb{R}^{B_r \times B_c}$.
30:         Accumulate $f(\tau_i), f'(\tau_i), f''(\tau_i)$ using $P_i^{(j)}$.
31:         **if** $\text{any}(P_i^{(j)} > 0)$ **then**
32:             $\mathcal{M}_{ij} \leftarrow 1$.
33:         **end if**
34:     **end for**
35:     $\tau_i \leftarrow \text{HybridSolver}(f, f', f'') \in \mathbb{R}^{B_r}$         ▷ See §3.2
36:
37:     **// Phase 4: Compute output for nonzero blocks**
38:     Initialize $O_i \leftarrow 0_{B_r \times d}$ on SRAM.
39:     **for** $j : \mathcal{M}_{ij} = 1$ **do**
40:         Load $K_j, V_j$ from HBM to SRAM.
41:         Compute $S_i^{(j)} \leftarrow Q_i K_j^\top \in \mathbb{R}^{B_r \times B_c}$.
42:         Compute $P_i^{(j)} \leftarrow \max\big(0, \; (\alpha-1)S_i^{(j)} - \tau_i\big)^{1/\alpha-1} \in \mathbb{R}^{B_r \times B_c}$.
43:         $O_i \leftarrow O_i + P_i^{(j)} V_j \in \mathbb{R}^{B_r \times d}$.
44:     **end for**
45:     Write $O_i$ to HBM.
46:     Write $\tau_i$ to HBM.
47: **end for**
48: **Return:** $O$ (and saved $\tau, \mathcal{M}$).

---

---

**Algorithm 2** ADASPLASH-2 Backward Pass for $dK$ and $dV$

---

**Require:** Matrices $Q, K, V, O, dO \in \mathbb{R}^{n \times d}$ and binary mask $\mathcal{M} \in \mathbb{Z}^{\lceil n/B_r \rceil \times \lceil n/B_c \rceil}$ in HBM, vector $\tau \in \mathbb{R}^n$ in HBM, block sizes $B_c, B_r$, parameter $\alpha$. Assume previously computed $\delta_i = \sum_{j=1}^{n} S_{ij}^{2-\alpha} V_j / \|U_i\|_1$.

1: Divide $Q$ into $T_r = \lceil n/B_r \rceil$ blocks $Q_1, \ldots, Q_{T_r}$ of size $B_r \times d$ each, and divide $K, V$ into $T_c = \lceil n/B_c \rceil$ blocks $K_1, \ldots, K_{T_c}, V_1, \ldots, V_{T_c}$ of size $B_c \times d$ each.
2: Divide $dO$ into $T_r$ blocks $dO_1, \ldots, dO_{T_r}$ of size $B_r \times d$ each.
3: Divide $\tau$ into $T_r$ blocks $\tau_1, \ldots, \tau_{T_r}$ of size $B_r$ each.
4: Initialize and divide $dK, dV \in \mathbb{R}^{n \times d}$ into $T_c$ blocks $dK_1, \ldots, dK_{T_c}$ and $dV_1, \ldots, dV_{T_c}$ of size $B_c \times d$ each.
5: Divide $\delta$ into $T_r$ blocks $\delta_1, \ldots, \delta_{T_r}$ of size $B_r$.
6: **for** $1 \le j \le T_c$ **do**
7:     Load $K_j, V_j$ from HBM to on-chip SRAM.
8:     Initialize $dK_j = 0_{B_c \times d}$ on SRAM.
9:     Initialize $dV_j = 0_{B_c \times d}$ on SRAM.
10:     **for** $i : \mathcal{M}_{ij} = 1$ **do**
11:         Load $Q_i, dO_i, \tau_i, \delta_i$ from HBM to on-chip SRAM.
12:         On chip, compute $S_i^{(j)} = Q_i K_j^\top \in \mathbb{R}^{B_r \times B_c}$.
13:         On chip, compute $P_i^{(j)} = \max(0, (\alpha - 1) S_i^{(j)} - \tau_i)^{1/\alpha - 1} \in \mathbb{R}^{B_r \times B_c}$.
14:         On chip, compute $dV_j \leftarrow dV_j + (P_i^{(j)})^\top dO_i \in \mathbb{R}^{B_c \times d}$.
15:         On chip, compute $dP_i = dO_i V_j^\top \in \mathbb{R}^{B_r \times B_c}$.
16:         On chip, compute $U_i^{(j)} = P_i^{(j)^{2-\alpha}} \in \mathbb{R}^{B_r \times B_c}$.
17:         On chip, compute $dS_i^{(j)} = U_i^{(j)} \odot (dP_i^{(j)} - \delta_i) \in \mathbb{R}^{B_r \times B_c}$.
18:         On chip, compute $dK_j \leftarrow dK_j + (dS_i^{(j)})^\top Q_i \in \mathbb{R}^{B_c \times d}$.
19:     **end for**
20:     Write $dK_j, dV_j$ to HBM.
21: **end for**
22: **Return:** Gradients $dK, dV$.

---

**Algorithm 3** ADASPLASH-2 Backward Pass for $dQ$

---

**Require:** Matrices $Q, K, V, O, dO \in \mathbb{R}^{n \times d}$ and binary mask $\mathcal{M} \in \mathbb{Z}^{\lceil n/B_r \rceil \times \lceil n/B_c \rceil}$ in HBM, vector $\tau \in \mathbb{R}^n$ in HBM, block sizes $B_c, B_r$, parameter $\alpha$. Assume previously computed $\delta_i = \sum_{j=1}^{n} S_{ij}^{2-\alpha} V_j / \|U_i\|_1$.

1: Divide $Q$ into $T_r = \lceil n/B_r \rceil$ blocks $Q_1, \ldots, Q_{T_r}$ of size $B_r \times d$ each, and divide $K, V$ into $T_c = \lceil n/B_c \rceil$ blocks $K_1, \ldots, K_{T_c}, V_1, \ldots, V_{T_c}$ of size $B_c \times d$ each.
2: Divide $dO$ into $T_r$ blocks $dO_1, \ldots, dO_{T_r}$ of size $B_r \times d$ each.
3: Divide $\tau$ into $T_r$ blocks $\tau_1, \ldots, \tau_{T_r}$ of size $B_r$ each.
4: Initialize $dQ$ in HBM and divide it into $T_r$ blocks $dQ_1, \ldots, dQ_{T_r}$ of size $B_r \times d$ each.
5: Divide $\delta$ into $T_r$ blocks $\delta_1, \ldots, \delta_{T_r}$ of size $B_r$ each.
6: **for** $i = 1$ to $T_r$ **do**
7:     Load $Q_i, dO_i, \delta_i, \tau_i,$ from HBM to on-chip SRAM
8:     Initialize $dQ_i = 0_{B_c \times d}$ on SRAM.
9:     **for** $j : \mathcal{M}_{ij} = 1$ **do**
10:         On chip, compute $S_i^{(j)} = Q_i K_j^\top \in \mathbb{R}^{B_r \times B_c}$.
11:         On chip, compute $P_i^{(j)} = \max(0, (\alpha - 1) S_i^{(j)} - \tau_i)^{1/\alpha - 1} \in \mathbb{R}^{B_r \times B_c}$.
12:         On chip, compute $dP_i = dO_i V_j^\top \in \mathbb{R}^{B_r \times B_c}$.
13:         On chip, compute $U_i^{(j)} = P_i^{(j)^{2-\alpha}} \in \mathbb{R}^{B_r \times B_c}$.
14:         On chip, compute $dS_i^{(j)} = U_i^{(j)} \odot (dP_i^{(j)} - \delta_i) \in \mathbb{R}^{B_r \times B_c}$.
15:         On chip, compute $dQ_i \leftarrow dQ_i + dS_i^{(j)} K_j \in \mathbb{R}^{B_r \times d}$.
16:     **end for**
17:     Write $dQ_i$ to HBM
18: **end for**
19: **Return:** Gradient $dQ$

---

