# OpenReview forum: "AdaSplash-2: Faster Differentiable Sparse Attention"
_ICML.cc/2026/Conference — ICML 2026 regular_

### Official Review · Reviewer_aRoP · 2026-03-04

**Soundness:** 3
**Presentation:** 3
**Significance:** 4
**Originality:** 3
**Overall Recommendation:** 5
**Confidence:** 4

**Summary:**

The paper presents ADASPLASH-2, a hardware-aware implementation of $\alpha$-entmax attention with a \textbf{histogram-based initialization} for the entmax normalization threshold ($\tau$), designed to reduce iterative root-finding cost and improve training throughput on GPU. The key idea is to construct a compact histogram in on-chip SRAM and use it to obtain a lower-bound estimate of the exact threshold, followed by a safeguarded refinement step and sparse block skipping in forward/backward passes.

**Compliance With Llm Reviewing Policy:**

Affirmed.

**Final Justification:**

I am satisfied with their responses.

**Key Questions For Authors:**

Although the experimental results are promising, could you provide a more detailed breakdown of the trade-off between algorithmic complexity and actual performance improvements, particularly as block sparsity increases?

**Limitations:**

Grammar error; suggested revision for clarity.

**Strengths And Weaknesses:**

Paper Strengths
The paper addresses a significant bottleneck: the high computational cost of $\alpha$-entmax normalization, which hinders efficient training throughput for competitive sparse differentiable attention mechanisms. Histogram-in-SRAM initialization is a practical and elegant idea that directly targets root-finding passes.

Major Weaknesses
1. What is the exact delta vs ADASPLASH?  Why is FA-3 omitted entirely from empirical plots? What is the memory overhead of histogram + masks vs FA-2 across contexts?
2. The proposed method shows promising speedups, but increased computational complexity for longer sequences or dense attention could be significant. The need for multiple passes to compute the threshold (τ) may also add overhead, especially for large models.
3. ADASPLASH-2 optimization appears tailored for NVIDIA GPUs, limiting its use on other hardware platforms. The method's performance on TPUs or multi-node systems is not clearly addressed.
4.  L210–212: ”outperforms a highly-optimized CUDA implementation...”  needs clarification. Name benchmark families for clarity.  L363–365: Clarify overbroad statement regarding α-entmax.   L119: ”The key operation requiring efficient implementations is normalization.” Suggested: The key operation requiring an efficient implementation is normalization.

---

> ### Author Rebuttal · Authors · 2026-03-31
>
> We thank the reviewer for the thoughtful review and constructive suggestions. We address your points below.
>
> > “What is the exact delta vs ADASPLASH?”
>
> The main improvement over AdaSplash-1 is not only an implementation refinement, but a different normalization strategy combined with a lighter sparsity mechanism. Concretely, AdaSplash-2 introduces (i) a histogram-based initialization for the entmax threshold, built directly from streamed scores in on-chip SRAM, reducing the number of refinement passes, (ii) a new hybrid solver that combines Halley, Newton, Secant and Bisection (rather than just Halley-Bisection), with theoretical convergence guarantees, and (iii) a lightweight bitpacked block mask that makes skipping zero blocks substantially cheaper.
>
> In the experimental side, we include controlled runtime comparisons against AdaSplash-1 in the efficiency section, where AdaSplash-2 is consistently faster under matched settings across sparsity levels and context lengths (Figs. 1 and 4). In addition, the solver analysis in Fig. 3 shows that histogram initialization places the threshold estimate much closer to the true root, which directly explains the runtime gain.
>
> > “Why is FA-3 omitted entirely from empirical plots?”
>
> As stated in the beginning of Section 4.2, our efficiency benchmarks target NVIDIA Ampere GPUs, for which FA2  is the relevant softmax reference. FA3 introduces Hopper-specific optimizations such as TMA and warp specialization, while preserving the same high-level softmax attention algorithm presented in FA2. Our goal was therefore to isolate the algorithmic contribution of AdaSplash-2 from Hopper-specific low-level kernel engineering.
>
> > “What is the memory overhead of histogram + masks vs FA-2 across contexts?”
>
> The added persistent memory overhead in AdaSplash-2 is dominated by the bitpacked block mask as the histogram itself is constructed in SRAM and does not require storing additional tensors in global memory.
>
> For sequence length $L$ and block size $b$, the block mask requires $(L/b)^2$ bits per head. For example, at 32K context with block size 64, this corresponds to 32 KB per head, which is negligible for current GPUs. We will add this formula together with concrete examples across context lengths in the revision.
>
>
> > “The proposed method shows promising speedups, but increased computational complexity for longer sequences or dense attention could be significant. The need for multiple passes to compute the threshold (τ) may also add overhead, especially for large models.”
>
> Indeed, AdaSplash-2 does incur additional work in the forward pass because computing the entmax normalization threshold requires extra passes over the keys. Accordingly, we do not expect it to outperform dense softmax attention in low-sparsity regimes. The key point, however, is that AdaSplash-2 benefits from increasing block sparsity, and empirically this sparsity grows with context length. As a result, the savings from skipping zero blocks increasingly compensate for the normalization overhead, especially in the backward pass, where the runtime gains are largest. This crossover behavior is precisely what is reflected in Figs. 1 and 4: at low sparsity, FlashAttention-2 remains favorable, while at moderate-to-high sparsity AdaSplash-2 becomes competitive and eventually faster.
>
> Given the importance of the passes over key blocks, we again re-stress the importance of reducing the number of the solver iterations in Fig. 3. That figure highlights a major improvement of AdaSplash-2 over the simple solver of AdaSplash (green line).
>
> > “ADASPLASH-2 optimization appears tailored for NVIDIA GPUs, limiting its use on other hardware platforms. The method's performance on TPUs or multi-node systems is not clearly addressed.”
>
> While our empirical claims in the paper are limited to the NVIDIA GPUs, the underlying ideas of histogram-based initialization and lightweight block masking are not conceptually tied to one specific accelerator. In fact, since our implementation is written in Triton, one can also target AMD GPUs.
>
> > “Although the experimental results are promising, could you provide a more detailed breakdown of the trade-off between algorithmic complexity and actual performance improvements, particularly as block sparsity increases?”
>
> Our method improves performance not by reducing the fixed cost of attention universally, but by changing the trade-off between an additional normalization overhead and the ability to avoid work on zero blocks. Concretely, AdaSplash-2 incurs extra forward-pass cost from histogram construction and threshold refinement, which makes the dense or low-sparsity regime less favorable. However, this overhead is approximately constant, whereas the computational savings grow with block sparsity because more blocks can be skipped. This is why the speedups become larger as sparsity increases, with the strongest gains appearing in the backward pass and in moderate-to-high sparsity settings.

---

> > ### Author Rebuttal · Reviewer_aRoP · 2026-04-02
> >
> > I thank the authors for the additional discussion and for addressing my comments in detail. I am satisfied with their responses.

---

### Official Review · Reviewer_QgBe · 2026-03-10

**Soundness:** 2
**Presentation:** 3
**Significance:** 3
**Originality:** 2
**Overall Recommendation:** 4
**Confidence:** 2

**Summary:**

The paper proposes ADASPLASH-2, a faster implementation of α-entmax attention that combines a histogram-based initialization for threshold estimation with sparsity-aware GPU kernels. The author argues this substantially improves the practicality, shows favorable training-time comparisons against prior ADASPLASH and FlashAttention-2 under sufficiently high block sparsity, and reports strong long-context results, especially for α-entmax+NAPE.

**Compliance With Llm Reviewing Policy:**

Affirmed.

**Final Justification:**

The additional clarification on Hopper-side efficiency, the added fixed-sparse baseline, and the explanation of the NAPE vs RoPE behavior address several of my main concerns. I still believe some limitations remain. In particular, while the paper appropriately scopes its efficiency claims primarily to training, this also means that the practical benefits for inference-time deployment remain unverified. I also think broader fixed-sparse and larger-scale comparisons would strengthen the paper. However, the rebuttal resolves enough of my concerns that I am comfortable updating my score.

**Key Questions For Authors:**

1. Could you comment on whether α-entmax  could be replaced or approximated after training by a faster inference-time mechanism, given that forward remains slower than FlashAttention-2 and inference is entirely forward-pass driven?

2. Could you provide timing comparisons against FlashAttention-2 on Hopper hardware? Such results would help clarify whether the reported efficiency gains transfer to the hardware regime used in the main experiments.

3. Could you explain why α-entmax + NAPE performs substantially better than the other variants, while α-entmax + RoPE does not, and what this suggests about the interaction between α-entmax sparsity and positional encoding?

**Limitations:**

Yes

**Strengths And Weaknesses:**

**Strengths:**

1. The paper presents a clear improvement over prior ADASPLASH through a stronger threshold-initialization strategy and a more efficient sparse kernel design, addressing a real bottleneck in making α-entmax practical.

2. The paper evaluates both efficiency and language modeling quality, including long-context benchmarks such as RULER and HELMET ICL, which makes the empirical study reasonably broad.

3. The paper shows that α-entmax+NAPE can outperform softmax baselines on long-context benchmarks, suggesting that sparse attention may be beneficial.


---


**Weaknesses:**

1. I found it somewhat inconsistent that the paper trains its LMs on 4×H100 GPUs but reports all microbenchmarks on a single A6000. While this benchmarking choice may help isolate algorithmic improvements from Hopper-specific engineering effects, it remains unclear whether the reported speedups would persist on the hardware used for the paper’s language-model training experiments.

2. A central motivation of the paper is that α-entmax provides input-dependent dynamic sparsity, implicitly suggesting an advantage over fixed sparse attention patterns. However a direct comparison against fixed block-sparse attention baselines, either in terms of downstream performance or compute efficiency is missing.

3. A practically important limitation is that ADASPLASH-2 remains noticeably slower than FlashAttention-2 in the forward pass. This matters because, while the method may improve training efficiency through backward-pass savings, inference is entirely forward-pass driven and is often the dominant deployment cost in practice. The paper’s benefits appear to be mainly training-side, which limits the broader practical impact.

4. The evaluation is limited to 350M and 1B models, and the trend appears scale-dependent. For example, Table 3 suggests that under RoPE, α-entmax is modestly competitive at 350M but weaker relative to softmax at 1B. This makes the overall advantage harder to interpret and suggests that an additional model scale would be useful.

5. Minor typo: In Section 4.2 (Context Scaling), “constrasting” should be “contrasting.”

---

> ### Author Rebuttal · Authors · 2026-03-31
>
> Thank you for the careful review. We address your points below.
>
> > “…However a direct comparison against fixed block-sparse attention baselines, either in terms of downstream performance or compute efficiency is missing.”
>
> To address the reviewer’s concern, we trained a softmax sliding-window baseline model (1B) with window size 512 and NAPE positional encoding, and evaluated it on RULER at 4K against the entmax model. We find a large gap in favor of entmax+NAPE: 41.8% avg. for entmax attention versus 10.3% for sliding-window softmax. This suggests that the gains are not explained by sparsity alone, but by the ability to adapt the sparsity to the input, rather than committing to a fixed local pattern.
>
> > “A practically important limitation is that ADASPLASH-2 remains noticeably slower than FlashAttention-2 in the forward pass.”
>
> AdaSplash-2 is a training-oriented method, so the main practical benefit we target is to lower end-to-end training step time, driven in particular by backward-pass savings from sparsity.
>
> We do not believe the slower forward pass eliminates its broader practical relevance. Even though this is not the focus of the current paper, our method could potentially be adapted for inference settings. Decoding is a distinct systems problem, and a training-oriented block kernel is not the right implementation target there. In our measurements, token-level sparsity at inference time is substantially higher than the block sparsity seen during training (approximately 87/90/93/95% at 4K/8K/16K/32K, averaged across layers and heads for the 1B entmax+NAPE model), which suggests substantial room for a decoding-specific kernel that exploits finer-grained sparsity.
>
> We provide further discussion on this point in our response to Reviewer j2HU, point 2.
>
> > “The evaluation is limited to 350M and 1B models, and the trend appears scale-dependent. For example, Table 3 suggests that under RoPE, α-entmax is modestly competitive at 350M but weaker relative to softmax at 1B. ”
>
> Our goal is to show that AdaSplash-2 makes entmax attention practical and competitive; for that purpose, 350M and 1B already provide meaningful robustness tests. At the same time, we also do not think the results indicate a generic degradation with scale. Rather, the apparent scale dependence is tied primarily to the positional encoding choice: the weaker trend is concentrated in the RoPE setting, whereas entmax + NAPE is the strongest overall variant in our long-context experiments (in both RULER and HELMET).
>
> > “Could you comment on whether α-entmax could be replaced or approximated after training by a faster inference-time mechanism, given that forward remains slower than FlashAttention-2 and inference is entirely forward-pass driven?”
>
> This is a great question. As stated previously, during inference, we believe that a dedicated kernel could exploit finer-grained sparsity at a token-level (which is extremely high, e.g., >95% at 32k) more effectively than the current block-based design. As for your suggestion, we believe the early stages used to estimate $\tau$ could likely be run in lower precision, while the final refinement and output computation remain in the target precision. These are interesting directions for future work, and we thank the reviewer for the suggestion.
>
> > “Could you explain why α-entmax + NAPE performs substantially better than the other variants, while α-entmax + RoPE does not, and what this suggests about the interaction between α-entmax sparsity and positional encoding?”
>
> Recent work [1] suggests that RoPE is a poor match for dynamic sparse attention. Because entmax produces exact zeros, it is highly sensitive to the shape of the attention logits; under RoPE, the oscillatory relative-position structure can turn into fragmented, unstable sparse support, especially at long context. By contrast, NAPE appears better aligned with sparse attention: its ALiBi component provides a monotonic distance bias that acts like a “dynamic sliding window”, while its NoPE component preserves flexibility for content-based attention. Empirically, this combination appears to produce sparsity patterns that are more stable and generalize better, consistent with previous findings [1].
>
> > “I found it somewhat inconsistent that the paper trains its LMs on 4×H100 GPUs but reports all microbenchmarks on a single A6000.”  / “Could you provide timing comparisons against FlashAttention-2 on Hopper hardware?”
>
> We thank the reviewer for raising these points. These concerns overlap with the related questions raised by Reviewer WzYZ (Points 1 and 4). We refer the reviewer to our response to those points.
>
> [1] Vasylenko  et al. "Long-context generalization with sparse attention." ICLR 2026.

---

> > ### Author Rebuttal · Reviewer_QgBe · 2026-04-02
> >
> > Thank you for the detailed rebuttal. The additional clarification on Hopper-side efficiency, the added fixed-sparse baseline, and the explanation of the NAPE vs RoPE behavior address several of my main concerns. I still believe some limitations remain. In particular, while the paper appropriately scopes its efficiency claims primarily to training, this also means that the practical benefits for inference-time deployment remain unverified. I also think broader fixed-sparse and larger-scale comparisons would strengthen the paper. However, the rebuttal resolves enough of my concerns that I am comfortable updating my score.

---

### Official Review · Reviewer_j2HU · 2026-03-12

**Soundness:** 4
**Presentation:** 3
**Significance:** 3
**Originality:** 3
**Overall Recommendation:** 5
**Confidence:** 4

**Summary:**

This paper proposes a novel algorithm for computing sparse entmax attention. Building on a prior algorithm, the authors introduce a preprocessing step to reduce the number of iterations required by the main root-finding algorithm to 1-2 iterations. The authors demonstrate significant speedup over the prior implementation, making entmax attention more competitive with standard softmax attention implementations (i.e. FlashAttention).

**Compliance With Llm Reviewing Policy:**

Affirmed.

**Final Justification:**

As per my review, this is an overall well-executed paper that presents a clear improvement over previous implementations of sparse entmax attention, hopefully increasing its adoption over softmax attention. I recommend acceptance.

**Key Questions For Authors:**

- Can the ideas presented here be extended to decoding inference? If so, how would it compare with top-k?
- We know that Newton's method can converge quadratically when close enough to the solution - would it be possible to find this interval for the given root-finding objective, and then use this to choose the bin sizes? It would be interesting to have some guarantee on when a single iteration is sufficient for refinement.

**Limitations:**

yes

**Strengths And Weaknesses:**

This is a very technically sound paper. The ideas are well-motivated by hardware considerations, and theoretical justification is provided for the histogram preprocessing step. Altogether, the ideas culminate in significant speedups over the previous implementation. The paper is overall well-organized and easy to read, although it might be helpful to have an algorithm listing that shows everything in one place (perhaps in the appendix if it takes too much space). Although entmax mappings are not mainstream in current Transformer implementations, this paper takes a great step in making entmax attention more attractive and perhaps improves its adoption by the community. While the main innovation can be seen as a "smart" initialization, the discretization step is quite inspired and novel as far as I know.

---

> ### Author Rebuttal · Authors · 2026-03-30
>
> Thank you for the positive assessment and for the thoughtful technical questions. We are glad that you  found the discretized histogram initialization both novel and well-motivated from the hardware perspective. We address your points below.
>
> > “... it might be helpful to have an algorithm listing that shows everything in one place ...”
>
> We provide the detailed forward and backward procedures in Appendix F. We will make sure to make this easier to locate and add a compact summary listing at the beginning of the appendix in the final version.
>
> > “Can the ideas presented here be extended to decoding inference? If so, how would it compare with top-k?”
>
> We believe the core ideas, such as histogram-based threshold initialization and lightweight block masking, are compatible with decoding inference, but a high-performance decoding kernel is a distinct systems problem from the training setting studied in this paper. In particular, decoding places the computation in a different, highly memory-bound regime, so a naive port of the training kernel would likely underutilize the GPU. At the same time, in our measurements, token-level sparsity at inference time is substantially higher than the block sparsity we report during training (approximately 87/90/93/95% sparse at 4K/8K/16K/32K, averaged across layers and heads for the 1B entmax+NAPE model), which suggests substantial room for a decoding-specific kernel that exploits finer-grained sparsity. Our current implementation is explicitly training-oriented, and extending it to decoding would require a separate kernel design and evaluation setup, as is also the case for softmax attention (e.g., FlashDecoding-style kernels). For this reason, we have deliberately scoped our claims to training-time efficiency. We will further clarify this scope in the revision.
>
> Regarding top-k, we view it as complementary rather than a strict comparison. Top-k imposes sparsity through explicit selection, while entmax induces sparse support through the normalization itself and remains end-to-end differentiable during training. In practice, top-k is often attractive for decoding because it gives direct control over compute, whereas entmax is appealing as an intrinsically sparse transformation. One could also combine the two by using top-k as a candidate-pruning step before applying entmax, but that would define a hybrid approximate variant rather than exact full entmax attention. We therefore prefer not to make unsupported claims about relative decoding performance without a dedicated inference kernel and evaluation.
>
>
> > “We know that Newton's method can converge quadratically when close enough to the solution - would it be possible to find this interval for the given root-finding objective, and then use this to choose the bin sizes?”
>
> Thank you for the insightful suggestion. In principle, one could characterize a neighborhood around the true root in which Newton-type can converge quadratically, and use that to design adaptive histogram bins.
> In the paper, we opted for a simpler distribution-agnostic guarantee on the initialization error together with the empirical observation that histogram initialization already places the solver orders of magnitude closer to the true root, making refinement effectively a 1 to 2 step procedure in practice.
> From a systems perspective, this choice is also more hardware-friendly, since fixed binning is simple to implement efficiently on GPU, whereas adaptive bin sizing would introduce additional complexity.

---

> > ### Author Rebuttal · Reviewer_j2HU · 2026-04-03
> >
> > Thank you for the insightful answers, I maintain my positive assessment of the paper.

---

### Official Review · Reviewer_WzYZ · 2026-03-12

**Soundness:** 3
**Presentation:** 3
**Significance:** 2
**Originality:** 3
**Overall Recommendation:** 4
**Confidence:** 3

**Summary:**

The authors propose AdaSplash-2, an efficient sparse attention framework based on α-entmax, targeting long-context LLM training. The key contribution is a histogram-based normalization scheme that builds a compact histogram of attention scores in on-chip SRAM, greatly reducing required iterations. The author implemented AdaSplash-2 in Triton and evaluated on synthetic efficiency benchmarks and language modeling tasks, reporting training speedups over FlashAttention-2 at moderate-to-high block sparsity and improved long-context benchmark scores relative to softmax baselines.

**Compliance With Llm Reviewing Policy:**

Affirmed.

**Final Justification:**

My final recommendation remains positive. The paper offers solid theoretical grounding and algorithm-hardware co-design for sparse attention. While I objectively understand the engineering complexity of an FA3-level implementation, FlashAttention-3 (and eventually FA4) represents a real-world performance barrier; if the practical efficiency gains of this sparse method over FA3 are limited, the overall impact of this work will be constrained in practice. Additionally, I still have reservations about the method's impact on training stability. It is crucial that the authors address this in the final version by providing comprehensive comparisons of the training loss curves.

**Key Questions For Authors:**

Q1. Can the authors provide training loss curves comparing the softmax baseline and the proposed method over the 50B training tokens to verify that the dynamic sparsity does not negatively affect convergence stability?

Q2. Could the authors provide a more representative estimate of practical training and inference throughput gains?

Q3.  The experiments do not compare directly against AdaSplash-1 under identical settings. Could the authors add a controlled comparison between AdaSplash-1 and AdaSplash-2 to isolate the contribution of the histogram initialization and establish the incremental gain of this work over its predecessor?

**Limitations:**

yes

**Strengths And Weaknesses:**

### Strengths
* **Solid Theoretical Grounding.** The author provides solid justification of the proposed histogram approximation. More, the authors provide mathematical bounds ensuring the estimated threshold never overestimates the true root.

* **Algorithm Hardware Co-design.** The algorithm effectively utilize native GPU insturctions for packing bits, effectly reduce the memory foorprint and reduce the memory bandwidth pressure.

* **Strong Evalution Results.** The evaluation on down-stream benchmarks demonstrates that models trained with given methods maintain strong performance.

### Weaknesses
* **Lack Efficiency Evidence.** AdaSplash-2's forward pass requires multiple sequential scans over K/V tiles, compared to FlashAttention-2's single fused pass. The paper provides no per-phase runtime breakdown and reports no MFU. More, all efficiency benchmarks are performed on a single A6000 (Ampere Arch) GPU, while language model training is conducted on H100 (Hopper Arch) GPUs. In addition, the context-scaling experiment uses sparsity from `a fixed transformer layer that exhibits high block sparsity` rather than averaging across all layers (Figure 7 shows the overall mean sparsity at 4K–16K context ranges from 0.48 to 0.56, well below the threshold at which AdaSplash-2 outperforms FlashAttention 2).

* **Insufficient Analysis of Training Dynamics.** The paper jumps directly to final benchmark scores but omits comparative training loss curves. It is crucial to demonstrate that it does not negatively impact the stability or convergence rate during the pretraining. Also,

---

> ### Author Rebuttal · Authors · 2026-03-31
>
> Thank you for your insightful comments and suggestions. We address your points below.
>
> > “... all efficiency benchmarks are performed on a single A6000 (Ampere Arch) GPU, while language model training is conducted on H100 (Hopper Arch) GPUs.”
>
> As noted in Sec. 4.2, we benchmark on Ampere because FA2 is the appropriate reference there, whereas FA3 adds Hopper-specific optimizations that are orthogonal to our algorithmic contribution. We use H100s for language-model training simply because they enable faster large-scale runs with larger batch sizes.
> We now also report H100 microbenchmark results against FA2 under the same increasing-sparsity setup. On H100, the triton version is the strongest FA2 baseline, so we use it as the reference. Although AS2 (AdaSplash-2) is slower in the dense regime, the same pattern holds on Hopper: as sparsity increases, it closes the gap, surpassing FA2 around 65% block sparsity. This shows that the efficiency gains transfer to the Hopper hardware used in our language-model experiments.
> |Method|0%|60%|70%|80%|90%|98%|
> |-|-|-|-|-|-|-|
> |FA2 (Triton)|**1.00x**|**1.00x**|1.00x|1.00x|1.00x|1.00x|
> |FA2 (CUDA)|0.74x|0.74x|0.74x|0.74x|0.74x|0.74x|
> |AS2|0.56x|0.94x|**1.05x**|**1.20x**|**1.40x**|**1.59x**|
>
> > “In addition, the context-scaling experiment uses sparsity from a fixed transformer layer that exhibits high block sparsity rather than averaging across all layers...”
>
> We use a fixed transformer layer because AS2’s runtime relative to FA2 is driven mainly by block sparsity. For similar sparsity levels, we observe similar normalized runtime across context lengths, so the purpose of this experiment is to isolate how the learned sparsity that emerges at longer contexts affects runtime. We will clarify in the main text that this figure is a learned-sparsity case study rather than a whole-model throughput estimate, and add the corresponding all-layer average analysis in the appendix.
>
> > “It is crucial to demonstrate that it does not negatively impact the stability or convergence rate during the pretraining.” and Q1. “Can the authors provide training loss curves comparing the softmax baseline and the proposed method over the 50B training tokens...?”
>
> This is a great question. In the revision, we will add training-loss curves over the 50B-token pretraining run, comparing the softmax baseline and AS2 under the same training setup (1B variants trained with a context length of 4k). For now, we include the following table containing smoothed loss values at some training steps:
>
> |Steps|5k|25k|50k|75k|100k|
> |-|-|-|-|-|-|
> |Softmax NAPE|3.379|2.983|2.883|2.850|2.732|
> |Entmax NAPE|3.407|2.981|2.879|2.845|2.724|
>
> The respective curves show stable optimization throughout training, without instability spikes. Moreover, AS2 consistently achieves lower training loss across the run. So, to answer your question more directly: dynamic sparsity does not negatively affect convergence rate or stability.
>
>
>
> > Q2. “Could the authors provide a more representative estimate of practical training and inference throughput gains?”
>
> For training, here is the end-to-end throughput against FA2 during a 350M-model pretraining run on a H100. We break the run into equal parts Beginning / Middle / End stages, and also report the context-extension phase, to show how the practical throughput evolves as the model progresses:
>
> |Norm. Throughput|Beg.(4k)|Mid(4k)|End(4k)|CtxExt(32k)|
> |-|-|-|-|-|
> |FA2 (Triton)|1.00x|1.00x|1.00x|1.00x|
> |FA2 (CUDA)|0.93x|0.93x|0.93x|0.85x|
> |AS1|0.81x|0.83x|0.84x|0.86x|
> |AS2|0.92x|0.94x|0.95x|0.98x|
>
> AS2 is faster than AS1 and already close to the strongest FA2 baseline early in training, with the gap narrowing further over training and context extension as block sparsity increases, while delivering stronger downstream and long-context performance.  Following that trend, we estimate that doing context extension with 64k ctx would eventually make AS2 faster than FA2 (Triton).
>
> Now, extending the benefits seen during training to inference is a largely distinct problem that typically requires a separate kernel design and evaluation setup, as seen in the softmax setting as well (FlashAttention vs FlashDecoding). We view this as an interesting direction for future work and will clarify this scope in the revision.
>
> We also refer the reviewer to our response to Reviewer j2HU’s second point, where we discuss inference-time considerations in more detail.
>
>
> > Q3. “The experiments do not compare directly against AdaSplash-1 under identical settings...”
>
> This comparison is already included in Sec. 4.1 and 4.2. Under matched settings, AS2 is consistently faster than AdaSplash-1 across sparsity levels and context lengths (Figs. 1 and 4). Figure 3 also shows that histogram initialization places the threshold estimate much closer to the true root, reducing refinement steps and required key passes relative to AdaSplash-1 (green/Hybrid). We will make this comparison more explicit in the final version.

---

> > ### Author Rebuttal · Reviewer_WzYZ · 2026-04-02
> >
> > Thanks for the timely reply.

---

### Decision · Program_Chairs · 2026-04-30

**Decision:**

Accept (regular)

**Comment:**

The reviewers were consistently positive, and I agree that this paper should be accepted. In my view, the most important aspect of the work is not only the systems improvement, but the fact that it makes a sparse differentiable attention operator materially more plausible for real LLM training. Although α-entmax has not seen broad adoption so far, the paper provides compelling evidence that it can deliver meaningful accuracy gains over softmax-based models in long-context language modeling, while now being trainable efficiently enough to matter in practice. In particular, the long-context results on both RULER and HELMET are quite encouraging: the paper reports that α-entmax with NAPE achieves the strongest average performance on RULER across sequence lengths, outperforming both softmax baselines, and Table 2 also shows small but consistent improvements on HELMET-ICL relative to the corresponding softmax settings.

I therefore view this paper as more than implementation refinement over prior AdaSplash-style work. The main contribution is that it substantially lowers the computational barrier that has historically limited the use of entmax attention, while pairing that efficiency improvement with empirical evidence that the operator may offer genuine modeling advantages in long-context settings.

At the same time, there are limitations that should be acknowledged clearly. The current evidence is still restricted to dense LLaMA-3 style models at 350M and 1B scale trained for 50B tokens; the paper does not evaluate MoE settings, nor does it yet provide an efficient decode-time kernel, which leaves the serving story incomplete. I do not see these as blocking concerns for acceptance, but rather as the main next steps needed to strengthen the case for broad deployment.

Overall, I find the paper technically sound, well-motivated, and useful. It advances the practicality of α-entmax attention in an important way, and the reported long-context gains make the work potentially consequential for future LLM training research. I therefore recommend acceptance.